# Diffusion-Classifier Synergy: Reward-Aligned Learning via Mutual Boosting Loop for FSCIL

**Ruitao Wu**[1,2]    **Yifan Zhao**[1]*    **Guangyao Chen**[3]    **Jia Li**[1]

[1]State Key Laboratory of Virtual Reality Technology and Systems, SCSE & QRI, Beihang University
[2]Zhongguancun Academy    [3]Peking University
{ruitaowu, zhaoyf}@buaa.edu.cn

## Abstract

Few-Shot Class-Incremental Learning (FSCIL) challenges models to sequentially learn new classes from minimal examples without forgetting prior knowledge, a task complicated by the stability-plasticity dilemma and data scarcity. Current FSCIL methods often struggle with generalization due to their reliance on limited datasets. While diffusion models offer a path for data augmentation, their direct application can lead to semantic misalignment or ineffective guidance. This paper introduces Diffusion-Classifier Synergy (DCS), a novel framework that establishes a mutual boosting loop between diffusion model and FSCIL classifier. DCS utilizes a reward-aligned learning strategy, where a dynamic, multi-faceted reward function derived from the classifier's state directs the diffusion model. This reward system operates at two levels: the feature level ensures semantic coherence and diversity using prototype-anchored maximum mean discrepancy and dimension-wise variance matching, while the logits level promotes exploratory image generation and enhances inter-class discriminability through confidence recalibration and cross-session confusion-aware mechanisms. This co-evolutionary process, where generated images refine the classifier and an improved classifier state yields better reward signals, demonstrably achieves state-of-the-art performance on FSCIL benchmarks, significantly enhancing both knowledge retention and new class learning.

## 1 Introduction

Class-Incremental Learning (CIL) [65, 74, 58, 81, 31, 78] endeavors to equip models with the ability to learn new classes sequentially without forgetting previously acquired knowledge, a critical capability for real-world, dynamic environments. Few-Shot Class-Incremental Learning (FSCIL) [53, 52, 77, 50, 76] intensifies this challenge by further constraining that new classes are introduced with only a handful of training examples. This exacerbates the notorious *stability-plasticity dilemma* [34] and introduces severe *unreliable empirical risk minimization* [61] for novel classes due to data scarcity. The core of these issues lies the model's restricted access to past data and the limited information available for new concepts [32].

A common thread among mainstream FSCIL approaches is their reliance on the limited knowledge encapsulated within the initially provided datasets. This inherent limitation hinders their ability to significantly enhance both *intra-task generalization* (robustness within a learned class) and *inter-task generalization* (adaptability and discrimination across incrementally learned classes). The advent of powerful generative models, especially diffusion models [10, 16, 44, 48], offers a promising avenue to overcome these data limitations by synthesizing additional training samples. By generating images for old classes, they can facilitate knowledge replay with-

---

*Corresponding authors.

39th Conference on Neural Information Processing Systems (NeurIPS 2025).

out explicit storage, and by augmenting new, few-shot classes, they can provide richer training signals. The strong generalization capabilities of pre-trained diffusion models suggest they can introduce diverse and novel variations, potentially improving classifier robustness.

However, the direct application of diffusion models for data generation in FSCIL is not without significant hurdles. As identified in our analysis (detailed in Section 4.1), two primary problems emerge: (i) **Semantic Misalignment and Diversity Deficiency of Generated Images** (Figure 1(a) ①②). When conditioned solely on class names, vanilla diffusion models tend to generate images with significant semantic deviations or insufficient diversity from the FSCIL dataset. This results in misrepresentative or overly concentrated distributions, which can introduce noise and distort learned decision boundaries, thereby degrading classifier performance. (ii) **Inefficient Feedback for Guiding Image Generation** (Figure 1(a) ③). Existing generative methods [2, 47] typically lack a mechanism for the diffusion model to adapt its output based on the classifier's current state or learning needs. The generation process is often *blind* to whether the synthesized samples are genuinely beneficial, too easy, or too difficult for the classifier, or whether they address critical areas of confusion in the feature space. While million-scale image generation performs well [4], its application is impractical in resource-demanding settings like FSCIL. We therefore prioritize the efficiency associated with generating fewer ($< 50$) images per class.

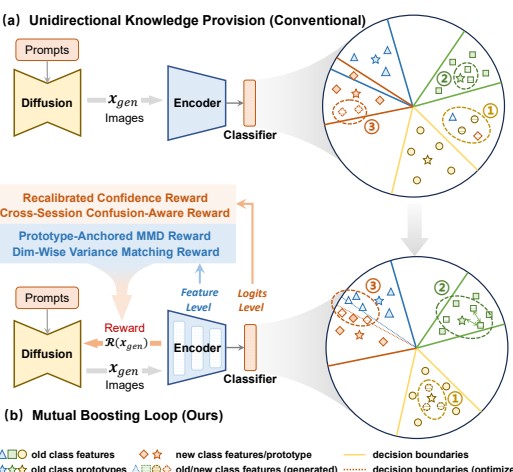

Figure 1: **(a)** Unidirectional knowledge provision in conventional methods results in inefficient generated images. **(b)** Our approach mitigates this inefficiency via a combined feature-level and logits-level reward, facilitated by a mutual boosting loop between the diffusion model and classifier.

To address these challenges, we introduce **Diffusion-Classifier Synergy (DCS)**, a novel framework that establishes a mutual boosting loop between diffusion model and FSCIL classifier. The core idea is to leverage the Diffusion Alignment as Sampling (DAS) [21] algorithm guided by a carefully designed, dynamic reward function. This reward, derived from the classifier's state, steers the diffusion model to generate strategically beneficial images. These generated images, in turn, enhance the classifier's training, leading to more refined reward signals, thus creating a co-evolutionary process.

Our DCS framework tackles the aforementioned issues at two distinct levels (Figure 1(b)): (i) At the **feature level**, where rewards are computed from features extracted by the encoder from images generated by the diffusion model, we introduce a reward design for semantic coherence and diversity (Section 4.2). This mechanism introduces two key rewards. The prototype-anchored maximum mean discrepancy reward function using MMD to encourage generated image diversity while maintaining consistency with class prototypes. Complementing this, the dimension-wise variance matching reward operates by aligning per-dimension feature variances of generated images with those from limited real data, offering a robust approach to match feature spread for new classes. This ensures that generated images are not only semantically anchored to the target class representations but also exhibit rich intra-class variations. (ii) At the **logits level**, where rewards are computed from classifier outputs for images generated by the diffusion model, we introduce a reward design for classifier-aware generation (Section 4.3). This design incorporates a recalibrated confidence reward to encourage the generation of more exploratory and generalized intra-class images, complementing the feature-level diversity reward. Building upon this, a novel cross-session confusion-aware reward is proposed, wherein the core idea is to intentionally generate *hard* samples that target the classifier's weaknesses. This is accomplished by adjusting the weight of classes in the cross-entropy based on the severity of their confusion, with the specific goal of enhancing discriminability between the target new class and its most confusable old classes.

Our contributions can be summarized as follows:

- We propose Diffusion-Classifier Synergy (DCS), a novel FSCIL framework that pioneers a mutual boosting loop between the diffusion model and the classifier, leveraging reward-aligned generation via DAS for synergistic co-evolution.

- We design a multi-faceted reward function operating at two distinct levels: at the feature level, it enforces semantic coherence and promotes intra-class diversity; and at the logits level, it guides the generation of exploratory, generalized intra-class images and enhances inter-class discriminability

- We empirically demonstrate state-of-the-art performance on challenging FSCIL benchmarks, showcasing significant improvements in both preserving knowledge of old classes and effectively learning new classes with limited data.

## 2 Related Works

**Class Incremental Learning**. In class-incremental learning tasks, models are required to continually learn to recognize new classes from a sequential data stream while retaining previously learned knowledge. Data replay-based methods [5, 23, 75, 59, 82] achieve this by storing data from previous tasks (such as image examples or features) or generating images of previously learned classes, allowing the model to revisit past data distributions. Network expansion-based methods [71, 64, 29, 65, 58, 79, 50] dynamically adjust the model's architecture or capacity during training to enhance its ability to learn new knowledge. Parameter regularization-based methods [69, 70, 25] focus on how the model parameters should dynamically adapt when the network structure remains fixed.

**Few-Shot Class-Incremental Learning**. FSCIL aims to achieve continual learning of new knowledge in data-constrained scenarios. Representative works such as [52, 73, 66, 67, 20, 38] focus on dynamic network-centric approaches, maintaining the topological relationships between feature spaces of various categories by dynamically adjusting the network structure. Methods like LIMIT [80, 8, 43] introduce the concept of meta-learning into FSCIL. Feature space-based methods [77, 41, 68, 51, 3, 27, 39, 37] enhance the robustness of the learned feature space by introducing virtual class instances and other means. Recently, many methods [40, 76, 17, 24] have employed pre-trained vision-language models (*e.g.*, CLIP) to further enhance the generalization capability of the models.

**Diffusion Models for Image Classification**. Data augmentation using diffusion model (DM) is an active research area aimed at enhancing image classifier training by synthesizing additional data. Key efforts focus on generating images that are both semantically consistent with class labels and exhibit sufficient richness and diversity to improve classifier performance. Initial studies [4, 35, 28, 13] demonstrated the viability of using fine-tuned DM for large-scale classification improvements, while subsequent research has explored advanced conditioning mechanisms [54] and image editing techniques [19, 62] to enhance semantic control and sample variety. Other approaches investigate leveraging DM features for precise guidance [30] or tailoring generation for specific scenarios like few-shot learning [56] or continual learning [46]. Distinct from the aforementioned approaches, our work focuses on exploring methodologies that dynamically adjust output content based on classifier feedback, with a particular emphasis on enhancing the efficiency of image generation in scenarios characterized by smaller generation scales (*e.g.*, tens of images rather than the conventional millions).

## 3 Preliminary

**Few-Shot Class-Incremental Learning** FSCIL involves sequentially learning from a continuous data stream $\mathcal{D}_{train} = \{\mathcal{D}_{train}^t\}_{t=0}^T$. Each session $t$ introduces training samples $(x_i, y_i)$ for a new set of disjoint classes $\mathcal{C}^t$. Crucially, after training on $\mathcal{D}_{train}^t$, the model is evaluated on all classes encountered thus far: $\mathcal{C}_{seen}^t = \bigcup_{s=0}^t \mathcal{C}^s$. FSCIL is characterized by an initial base session ($t = 0$) with ample data, followed by incremental sessions ($t > 0$) where new classes are introduced with only a few examples, typically in an $N$-way $K$-shot format.

Mainstream FSCIL methods tend to adopt an *incremental-frozen* strategy. An initial model $\sigma(x) = W^T f(x)$ (comprising a feature extractor $f$ and classifier $W$) is trained extensively on base session data using a classification loss, *e.g.*, cross-entropy:

$$\mathcal{L}_{cls}(\sigma; x, y) = \mathcal{L}_{ce}(\sigma(x), y). \tag{1}$$

Subsequently, the feature extractor $f$ is frozen. In incremental session $s$, the classifier weights $\boldsymbol{w}_c^s$ for new classes are typically computed as prototypes, which are defined as the average feature embeddings of their $K$ training samples, such that $\boldsymbol{w}_c^s = \frac{1}{K}\sum_{i=1}^K f(x_{c,i})$. The full classifier $W_{full}$

then encompasses weights for all seen classes. Inference at each session employs the Nearest Class Mean (NCM) algorithm [33]. A sample $x$ is classified by finding the class prototype most similar to its feature embedding $f(x)$: $c_x = \arg\max_{c,s} \text{sim}(f(x), \boldsymbol{w}_c^s)$, where $\text{sim}(\cdot, \cdot)$ is the cosine similarity.

**Diffusion Alignment as Sampling (DAS)**   Aligning pre-trained diffusion models with specific rewards while preserving their generative quality and diversity is a key challenge [9, 7, 12, 49, 42, 6, 14, 72]. Fine-tuning can lead to reward over-optimization, while simpler guidance methods may underperform. [21] proposed DAS, a *training-free* algorithm to address this. DAS aims to effectively sample from a reward-aligned target distribution without explicit model retraining, thereby mitigating over-optimization. The core of DAS is to sample from the target distribution $p_{tar}(x)$, which balances a reward function $r(x)$ with fidelity to the pre-trained model $p_{pre}(x)$:

$$P_{tar}(x) = \frac{1}{\mathcal{Z}} p_{pre}(x) \exp\left(\frac{r(x)}{\alpha}\right), \tag{2}$$

where $\mathcal{Z}$ is a normalization constant and $\alpha$ is a trade-off parameter. To achieve this, DAS employs Sequential Monte Carlo (SMC) [36]. It iteratively guides a set of particles (noisy samples) through the reverse diffusion process. A key feature of DAS is the use of tempered intermediate target distributions $\pi_t(x_t)$ at each diffusion timestep $t$:

$$\pi_t(x_t) \propto p_t(x_t) \exp\left(\frac{\lambda_t}{\alpha} \hat{r}(x_t)\right), \tag{3}$$

where $p_t(x_t)$ is the marginal distribution from the pre-trained model, $\hat{r}(x_t)$ is the predicted reward from noisy sample $x_t$, and $\lambda_t$ is an annealing schedule ($\lambda_T = 0$ to $\lambda_0 = 1$). Diverging from the DAS which predominantly utilized rewards such as HPSv2 [63] and TCE [18] to assess image quality, our approach, in order to specifically address the FSCIL task, involves inputting the generated image $\boldsymbol{x}_{\text{gen}}$ into the classifier. Rewards are subsequently computed based on the classifier's output, and this feedback signal is enhanced by combining multiple distinct rewards.

# 4   Reward-Aligned Learning via Mutual Boosting Loop for FSCIL

In this section, we introduce our novel framework, Diffusion-Classifier Synergy (DCS), which leverages a reward-aligned learning paradigm through a mutual boosting loop to address the inherent challenges of FSCIL. We first outline the primary issues in FSCIL and present the overarching workflow of our approach. Subsequently, we detail the design of our reward functions at both the feature and logits levels.

## 4.1   Addressing FSCIL Challenges with a Mutual Boosting Loop

FSCIL faces the *stability-plasticity dilemma* and *unreliable empirical risk minimization*. Generative models, particularly diffusion models, address this by synthesizing data: generating old class images for knowledge replay mitigates the stability-plasticity issue, while creating new class images augments data for few-shot learning. However, naively integrating diffusion models for data generation in FSCIL presents significant problems, which we will analyze in detail.

**Semantic Misalignment and Diversity Deficiency of Generated Images**. Synthesizing training images that balance semantic fidelity with background diversity presents a substantial challenge when generative prompts are restricted to class labels. More specifically, we employ Stable Diffusion to generate five images per *mini*ImageNet class across various guidance scales, subsequently extracting and classifying features using an ImageNet pre-trained ResNet34. Semantic alignment is determined by classification accuracy. In contrast, intra-class feature dispersion, an indicator of semantic richness quantified by the average L2 distance of features to their class centroid, shows an inverse correlation with accuracy as guidance is strengthened (Figure 2 Left). Critically, irrespective of guidance parameters, synthetic data consistently underperforms on both classification accuracy and semantic richness compared to ResNet34 on the original *mini*ImageNet testset (baseline).

**Inefficient Feedback Loop for Guiding Image Generation**. Mainstream dataset augmentation methods treat the diffusion model as a *blind* teacher, one that is unaware of the student classifier's needs, which hinders adaptive image generation in FSCIL. Specifically, it limits the ability to produce

samples tailored to the classifier's current capabilities, such as generating images near class decision boundaries. Such fine-grained control is crucial for alleviating semantic confusion between new and old classes. The t-SNE visualization in the right panel of Figure 2 indicates confusing regions between the two classes, which the classifier cannot sufficiently learn from the generated images, leading to easy misclassification of samples in this area.

To overcome these limitations, we propose the Mutual Boosting Loop. Its core idea is to design a multiple reward components $\mathcal{R}_i$, computed based on the output of the classifier $\sigma$ (with parameters $\theta$) when presented with a generated image $x$. The diffusion model $D$ is then guided to adjust its sampling strategy $\phi$ to maximize the sum of these rewards:

$$\phi^* = \arg\max_{\phi} \sum_i \mathcal{R}_i \left( \sigma_\theta(D(x; \phi)) \right). \quad (4)$$

The optimized $\phi^*$ guides image generation $D(x; \phi^*)$, which in turn enhances classifier performance:

$$\theta^* = \arg\min_{\theta} \mathcal{L}_{cls}(\sigma_\theta; x \cup D(x; \phi^*), y). \quad (5)$$

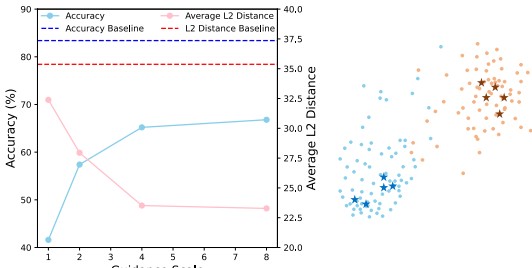

Figure 2: **Left:** Trade-off between semantic fidelity and richness with increasing guidance scale. **Right:** t-SNE visualization of features from real (dot) and generated (pentagram) images illustrates the restricted distribution of the latter within the classifier's decision space.

Consequently, the classifier, now optimized with parameters $\theta^*$, provides more accurate and informative reward signals back to the diffusion model, creating a synergistic co-evolution. The overall process is depicted in Section A. The subsequent sections will elaborate on the construction of this reward mechanism from feature-level (Section 4.2) and logits-level (Section 4.3) perspectives.

## 4.2 Feature-Level Reward for Semantic Coherence and Diversity

In this section, we detail the process for generating images of a specific class during each session, focusing on feature-level reward components. We assume the classifier has undergone initial learning, with its parameters (class prototypes) initialized using the few available samples for new classes. Addressing the first challenge identified in Section 4.1, a straightforward strategy is to enforce fidelity and diversity using the Fréchet Inception Distance (FID):

$$\text{FID} = \|\boldsymbol{\mu}_r - \boldsymbol{\mu}_g\|_2^2 + \text{Tr}\left(\boldsymbol{\Sigma}_r + \boldsymbol{\Sigma}_g - 2(\boldsymbol{\Sigma}_r \boldsymbol{\Sigma}_g)^{1/2}\right), \quad (6)$$

where $\boldsymbol{\mu}$ is the mean vector and $\boldsymbol{\Sigma}$ is the covariance matrix. Subscripts $r$ and $g$ indicate real and generated images, respectively. Despite its utility as an evaluation metric, directly using FID as a reward is impractical because restricted old data access prevents its calculation against the comprehensive dataset, and the limited number of new class samples hinders reliable $\boldsymbol{\Sigma}$ estimation, leading to unstable scores.

Drawing inspiration from the principles underlying FID, we propose a more suitable reward mechanism tailored to the constraints of FSCIL. Our goal is to achieve similar objectives of semantic coherence (related to mean matching) and feature diversity (related to covariance matching) using components that are robust with limited data. To approximate the goal of matching the overall feature distribution, particularly ensuring that generated images are semantically anchored to their class identity, we introduce the **Prototype-Anchored Maximum Mean Discrepancy Reward** $\mathcal{R}_{\text{PAMMD}}$. MMD is a statistical test used to determine if two sets of samples are drawn from the same distribution. Instead of directly comparing means which can be problematic with incomplete data for $\boldsymbol{\mu}_r$ and unstable $\boldsymbol{\mu}_g$ from few generated samples, MMD allows for a more holistic comparison.

When a candidate image $\boldsymbol{x}_{\text{gen}}$ is considered for addition to an existing set of $N-1$ generated images $\mathcal{I}_{\text{gen}}^{(c,N-1)}$ of class $y_c$ , $\mathcal{R}_{\text{PAMMD}}$ evaluates the quality of the augmented set $\mathcal{I}_{\text{gen}}^{(c,N)} = \mathcal{I}_{\text{gen}}^{(c,N-1)} \cup \{\boldsymbol{x}_{\text{gen}}\}$ by taking the negative of the MMD:

$$\mathcal{R}_{\text{PAMMD}}(\boldsymbol{x}_{\text{gen}}, \mathcal{I}_{\text{gen}}^{(c,N)}) = -\alpha \underbrace{\frac{1}{N^2} \sum_{i=1}^{N} \sum_{j=1}^{N} k(\boldsymbol{z}_i, \boldsymbol{z}_j)}_{\text{Diversity}} + \beta \underbrace{\frac{1}{N} \sum_{i=1}^{N} k(\boldsymbol{z}_i, \boldsymbol{\mu}_c)}_{\text{Consistency}} - \underbrace{k(\boldsymbol{\mu}_c, \boldsymbol{\mu}_c)}_{\text{Constant}}, \quad (7)$$

where $z_i = f(x_i)$ is the feature of the $i$-th generated image in $\mathcal{I}_{\text{gen}}^{(c,N)}$. $\boldsymbol{\mu}_c \in \mathbb{R}^D$ is the feature prototype for class $y_c$, which is in fact the class weight of the classifier. $k(\cdot, \cdot)$ is a positive definite kernel function, and $\alpha, \beta$ are non-negative hyperparameters.

The first term *Diversity* effectively measures the internal similarity of the $\mathcal{I}_{\text{gen}}^{(c,N)}$. A lower value for this component (contributing to a higher $\mathcal{R}_{\text{PAMMD}}$) is desirable, as it indicates that the generated images are distinct from each other, thereby maximizing diversity. The second term *Consistency* assesses the collective similarity of the generated images to the class prototype and encourages the $\boldsymbol{x}_{\text{gen}}$ to maintain strong semantic consistency with the target class. The third term is a constant that depends only on the class prototype, which can be ignored in the actual computation process. As diffusion models generate images sequentially, the gradient of the $\mathcal{R}_{\text{PAMMD}}$ is backpropagated solely to the currently generated image. To mitigate resource wastage from redundant computations, Equation (7) can be updated incrementally. The detailed procedure is provided in Section D.

A key advantage of $\mathcal{R}_{\text{PAMMD}}$ is its universality. It is applicable whether the model is generating images for previously learned classes or newly introduced classes. As the classifier continuously updates the knowledge acquired for the current class, the prototype $\boldsymbol{\mu}_c$ is also updated correspondingly, enabling it to provide more accurate feedback to the diffusion model in the subsequent session.

Despite the significant utility of $\mathcal{R}_{\text{PAMMD}}$, its reliance solely on prototypes when a subset of new-class images is accessible results in the underutilization of valuable information regarding the feature spread. Drawing inspiration from the second term of the FID, which utilizes covariance matrices to reflect the spatial distribution and spread of features, we propose the **Dimension-Wise Variance Matching Reward** $\mathcal{R}_{\text{VM}}$. This component is developed for robustly matching feature variances of novel classes where limited reference data is available and full covariance estimation is unstable.

We first estimate the per-dimension variances from the features extracted from real images $\mathcal{I}_{\text{real}}^{(c)}$. Then, for each new $\boldsymbol{x}_{\text{gen}}$, we evaluate how it affects the per-dimension variances of the $\mathcal{I}_{\text{gen}}^{(c,N)}$, and reward candidates that bring these variances closer to the target reference variances:

$$\mathcal{R}_{\text{VM}}(\boldsymbol{x}_{\text{gen}}, \mathcal{I}_{\text{gen}}^{(c,N)}) = -\sum_{d=1}^{D} \left(v_{\text{gen}}^d - v_{\text{real}}^d\right)^2, \tag{8}$$

where $v_{\text{gen}}^d = \text{Var}(\{f(\boldsymbol{x}_j)^d \mid \boldsymbol{x}_j \in \mathcal{I}_{\text{gen}}\})$ is the sample variance of the $d$-th dimension of features in $\mathcal{I}_{\text{gen}}$, and $v_{\text{real}}^d$ is defined analogously.

The rationale for dimension-wise matching, rather than full covariance matching, stems from the instability of covariance matrix estimation in few-shot scenarios. Estimating a $D \times D$ covariance matrix from very few samples (*e.g.*, $N \ll D$) can lead to highly inaccurate or singular matrices. Matching variances at the dimensional level alleviates this issue by focusing on more robust univariate statistics, though it forgoes capturing inter-dimensional correlations. Given that this matching process still entails certain sample size requirements, we recommend deferring the incorporation of this term into the overall reward until a sufficient number of images (*e.g.*, $> 5$) are generated.

## 4.3 Logits-Level Rewards for Robust Discrimination Learning

While feature-level rewards ensure semantic integrity and diversity, they do not directly leverage the classifier's decision-making process. To bridge this gap and enable finer-grained control over image generation, we introduce logits-level rewards that incorporate feedback from the current classifier.

A fundamental logits-level reward aims to encourage the generation of images that the classifier can correctly assign to the target class $y_c$. This can be initially conceptualized with a standard cross-entropy reward:

$$\mathcal{R}_{\text{CE}}(\boldsymbol{x}_{\text{gen}}, y_c) = \log(\hat{p}(y_c | \boldsymbol{x}_{\text{gen}})) = \log\left(\frac{\exp(\sigma_c(\boldsymbol{x}_{\text{gen}}))}{\sum_k \exp(\sigma_k(\boldsymbol{x}_{\text{gen}}))}\right), \tag{9}$$

where $\sigma_c(\boldsymbol{x}_{\text{gen}})$ is the logit for the target class $y_c$, $\hat{p}(y_c | \boldsymbol{x}_{gen})$ is the probability assigned by the current classifier to the target class $y_c$ for the generated image $\boldsymbol{x}_{\text{gen}}$. However, relying solely on maximizing this basic classification accuracy can lead to the generation of overly simplistic or peaky samples. To address this limitation and encourage the generation of more nuanced and informative samples, we

refine this concept into a **Recalibrated Confidence Reward** with temperature-scaled probability:

$$\mathcal{R}_{\text{RC}}(\boldsymbol{x}_{\text{gen}}, y_c) = \log(\hat{p}(y_c|\boldsymbol{x}_{\text{gen}}; T)), \tag{10}$$

$$\hat{p}(y_c|\boldsymbol{x}_{\text{gen}}; T) = \frac{\exp(\sigma_c(\boldsymbol{x}_{\text{gen}})/T)}{\sum_k \exp(\sigma_k(\boldsymbol{x}_{\text{gen}})/T)}, \tag{11}$$

$$T(\boldsymbol{x}_{\text{gen}}) = T_{base} + T_{scale} \cdot \left( \frac{\hat{p}_c(y_c|\boldsymbol{x}_{\text{gen}}) - 1/N_c}{1 - 1/N_c} \right), \tag{12}$$

where $N_c$ is the total class count. $T_{base} > 1$ provides baseline smoothing, and $T_{scale} > 0$ sets the maximum additional temperature. This adaptive temperature adjusts based on the classifier's raw confidence in the target class $y_c$ for $\boldsymbol{x}_{\text{gen}}$. A high $\hat{p}_c(y_c|\boldsymbol{x}_{\text{gen}})$ increases temperature $T$, flattening the reward's probability distribution to discourage overly simplistic samples. Conversely, a low $\hat{p}_c(y_c|\boldsymbol{x}_{\text{gen}})$ keeps $T$ near $T_{base}$, minimizing extra smoothing to prevent excessive diffusion. This efficient mechanism dynamically controls classifier feedback smoothness within the logits-level reward, thereby fostering the generation of more challenging and diverse samples.

While the $R_{RC}$ focuses on the confidence of the target class, it does not explicitly address the relationships between different classes. In FSCIL settings, a significant challenge arises when feature embeddings of new classes closely overlap with those of previously learned classes. Simply ensuring a high (even if smoothed) probability for the target class $y_c$ might not be sufficient to explicitly create a clear distinction from these confusing old classes. To this end, we introduce the **Cross-Session Confusion-Aware Reward** $\mathcal{R}_{CSCA}$, which makes it possible to generate *harder* samples for robust training and refine the classifier's understanding of nuanced class differences. To detail how this reward is formulated, we first outline how the classifier assesses these inter-class similarities and potential confusions.

The classifier's decision process involves computing the cosine similarity $\hat{s}(y_c|\boldsymbol{x}_{\text{gen}})$ between the features $f(\boldsymbol{x}_{\text{gen}})$ and the class prototype $\boldsymbol{\mu}_c$ for each class $y_c$, which is then used to derive the logits:

$$\hat{s}(y_c|\boldsymbol{x}_{\text{gen}}) = \frac{f(\boldsymbol{x}_{\text{gen}}) \cdot \boldsymbol{\mu}_c}{\|f(\boldsymbol{x}_{\text{gen}})\|\|\boldsymbol{\mu}_c\|}. \tag{13}$$

Based on this similarity, the cosine distance is $d_{\cos}(\boldsymbol{x}_{\text{gen}}, \boldsymbol{\mu}_c) = 1 - \hat{s}(y_c|\boldsymbol{x}_{\text{gen}})$, which quantifies how dissimilar the generated sample's features are from the class center $\boldsymbol{\mu}_c$. To modulate the influence of different classes based on their similarity to the generated sample, we introduce dynamic weights $w_{y_t}(\boldsymbol{x}_{\text{gen}})$ such that the weight for a class $y_t$ increases as the sample becomes more similar to the $\boldsymbol{\mu}_t$:

$$w_{y_t}(\boldsymbol{x}_{\text{gen}}) = \frac{1}{1 + \gamma \cdot d_{\cos}(\boldsymbol{x}_{\text{gen}}, \boldsymbol{\mu}_t)}, \tag{14}$$

where the scaling factor $\gamma > 0$ controls the sensitivity of the weight to the cosine distance. A smaller $d_{\cos}(\boldsymbol{x}_{\text{gen}}, y_t)$ result in more weight being assigned to the target class $y_t$ for the generated sample. Leveraging these weights, the $\mathcal{R}_{CSCA}$ is designed to encourage the generator to create samples for a target class $y_c$ that are highly similar to a confusable class $y_t$ within a set $\mathcal{C}$. This is achieved by defining the reward as the log-probability of classifying $\boldsymbol{x}_{\text{gen}}$ as $y_t$ instead of $y_c$, using dynamically weighted similarity scores as logits:

$$\mathcal{R}_{\text{CSCA}}(\boldsymbol{x}_{\text{gen}}, y_c) = \sum_{y \in \mathcal{C}} w_y(\boldsymbol{x}_{\text{gen}}) \log (\hat{p}(y|\boldsymbol{x}_{\text{gen}}; T_s)), \tag{15}$$

where $\mathcal{C}$ can be the set of the top-K most similar prototypes $\boldsymbol{\mu}_y$ to $\boldsymbol{x}_{\text{gen}}$ in order to reduce computation.

## 5 Experiments

### 5.1 Experimental Setup

**Datasets** Following the benchmark settings of previous methods, we conducted experiments on three datasets, *i.e.*, *mini*ImageNet [55, 45], CUB-200 [57], and CIFAR-100 [22]. The division of the datasets aligns with existing methods. Specifically, the CIFAR-100 and *mini*ImageNet datasets are partitioned into a base session containing 60 classes and incremental sessions containing 40 classes, with each session being an 8-way 5-shot few-shot classification task. The CUB-200 dataset is divided into the base session containing 100 classes and incremental sessions containing 40 classes, with each session being a 10-way 5-shot task.

Table 1: Comparison results on *mini*ImageNet dataset. * denotes results from [27].

| Methods | Accuracy in each session (%) ↑ | | | | | | | | | Avg ↑ |
|---|---|---|---|---|---|---|---|---|---|---|
| | 0 | 1 | 2 | 3 | 4 | 5 | 6 | 7 | 8 | |
| TOPIC [52] | 61.31 | 50.09 | 45.17 | 41.16 | 37.48 | 35.52 | 32.19 | 29.46 | 24.42 | 39.64 |
| CEC [73] | 72.00 | 66.83 | 62.97 | 59.43 | 56.70 | 53.73 | 51.19 | 49.24 | 47.63 | 57.75 |
| FACT [77] | 72.56 | 69.63 | 66.38 | 62.77 | 60.60 | 57.33 | 54.34 | 52.16 | 50.49 | 60.70 |
| TEEN [60] | 73.53 | 70.55 | 66.37 | 63.23 | 60.53 | 57.95 | 55.24 | 53.44 | 52.08 | 61.44 |
| SAVC [50] | 81.12 | 76.14 | 72.43 | 68.92 | 66.48 | 62.95 | 59.92 | 58.39 | 57.11 | 67.05 |
| DyCR [38] | 73.18 | 70.16 | 66.87 | 63.43 | 61.18 | 58.79 | 55.00 | 52.87 | 51.08 | 61.40 |
| ALFSCIL [26] | 81.27 | 75.97 | 70.97 | 66.53 | 63.46 | 59.95 | 56.93 | 54.81 | 53.31 | 64.80 |
| OrCo* [3] | **83.22** | 74.60 | 71.89 | 67.65 | 65.53 | 62.73 | 60.33 | 58.51 | 57.62 | 66.90 |
| ADBS* [27] | 81.40 | 75.03 | 71.03 | 68.00 | 65.56 | 61.87 | 59.04 | 56.87 | 55.38 | 66.02 |
| DCS(Ours) | 82.43 | **77.54** | **73.00** | **69.21** | **67.05** | **64.44** | **61.20** | **60.43** | **57.99** | **68.14** |

Table 2: Comparison results on CUB-200 dataset. * denotes results from [27].

| Methods | Accuracy in each session (%) ↑ | | | | | | | | | | | Avg ↑ |
|---|---|---|---|---|---|---|---|---|---|---|---|---|
| | 0 | 1 | 2 | 3 | 4 | 5 | 6 | 7 | 8 | 9 | 10 | |
| TOPIC [52] | 68.68 | 62.49 | 54.81 | 49.99 | 45.25 | 41.40 | 38.35 | 35.36 | 32.22 | 28.31 | 26.28 | 43.92 |
| CEC [73] | 75.85 | 71.94 | 68.50 | 63.50 | 62.43 | 58.27 | 57.73 | 55.81 | 54.83 | 53.52 | 52.28 | 61.33 |
| FACT [77] | 75.90 | 73.23 | 70.84 | 66.13 | 65.56 | 62.15 | 61.74 | 59.83 | 58.41 | 57.89 | 56.94 | 64.42 |
| TEEN [60] | 77.26 | 76.13 | 72.81 | 68.16 | 67.77 | 64.40 | 63.25 | 62.29 | 61.19 | 60.32 | 59.31 | 66.63 |
| SAVC [50] | 81.85 | **77.92** | **74.95** | 70.21 | **69.96** | 67.02 | **66.16** | 65.30 | 63.84 | 63.15 | 62.50 | 69.35 |
| DyCR [38] | 77.50 | 74.73 | 71.69 | 67.01 | 66.59 | 63.43 | 62.66 | 61.69 | 60.57 | 59.69 | 58.46 | 65.82 |
| ALFSCIL [26] | 79.79 | 76.53 | 73.12 | 69.02 | 67.62 | 64.76 | 63.45 | 62.32 | 60.83 | 60.21 | 59.30 | 67.00 |
| OrCo* [3] | 74.58 | 65.99 | 64.72 | 63.06 | 61.79 | 59.55 | 59.21 | 58.46 | 56.97 | 57.99 | 57.32 | 61.79 |
| ADBS* [27] | 79.99 | 75.89 | 72.53 | 68.33 | 67.92 | 64.75 | 64.10 | 62.93 | 61.31 | 60.88 | 59.65 | 67.12 |
| DCS(Ours) | **83.19** | 77.32 | 73.92 | **70.52** | 69.79 | **68.00** | 65.22 | **66.59** | **64.19** | **64.86** | **63.40** | **69.73** |

**Evaluation metrics**  Session accuracy quantifies model performance within a specific learning session. To evaluate sustained performance and generalization across both previously learned and newly introduced classes, average accuracy is calculated as the mean of all session accuracies from the initial to the current session.

**Implementation details**  For the Diffusion Alignment as Sampling (DAS) algorithm, we utilized the source code made available by the authors, into which our proposed reward mechanisms were integrated. The latest Stable Diffusion 3.5 Medium model [11] served as the foundational diffusion model. To align with the configuration of Stability AI's open-source weights, thereby ensuring the fidelity of the generated images, we adapted the DAS source code for compatibility with Flow Matching Scheduler. Images were generated at a resolution of $512 \times 512$ pixels and subsequently resized to match the native resolution of the real dataset before being input to the encoder. During the base session, an additional 30 images are generated per class. For new sessions, we generate 30 and 10 images for each newly introduced and previously learned class, respectively. For more details, please refer to Section B.

## 5.2 Comparison with the State of the Art

In this section, we compare our proposed DCS with mainstream methods on FSCIL benchmarks. Table 1 and Table 2 present the accuracy in each session and the average accuracy of these methods on the *mini*ImageNet and CUB-200, respectively.

Compared to previous state-of-the-art FSCIL methods, our proposed DCS achieves the highest accuracy in each session. Note that the compared methods employ network optimization techniques tailored to the characteristics of FSCIL tasks, including additional self-supervised learning or distribution calibration. In contrast, our DCS achieves performance improvement solely through the generalized knowledge derived from the diffusion model, without any modifications to the baseline classification network. For further results, please refer to Section C.

Table 3: Ablation studies on CIFAR-100 benchmark. $\Delta_{\text{last}}$: Relative improvements of the last sessions compared to the baseline.

| $\mathcal{R}_{\text{PAMMD}}$ | $\mathcal{R}_{\text{VM}}$ | $\mathcal{R}_{\text{RC}}$ | $\mathcal{R}_{\text{CSCA}}$ | Accuracy in each session (%) ↑ | | | | | | | | $\Delta_{\text{last}}$ |
|---|---|---|---|---|---|---|---|---|---|---|---|---|
| | | | | 1 | 2 | 3 | 4 | 5 | 6 | 7 | 8 | |
| | Baseline | | | 71.97 | 67.55 | 62.83 | 59.73 | 56.08 | 52.99 | 51.55 | 49.57 | - |
| ✓ | | | | 73.90 | 69.05 | 64.70 | 60.99 | 57.91 | 55.02 | 53.82 | 50.81 | +1.24 |
| ✓ | ✓ | | | 74.05 | 69.29 | 65.06 | 61.50 | 58.59 | 55.63 | 54.61 | 51.43 | +1.86 |
| ✓ | ✓ | ✓ | | 75.08 | 70.77 | 66.44 | 62.94 | 60.61 | 57.47 | 57.01 | 53.07 | +3.50 |
| ✓ | ✓ | ✓ | ✓ | **75.96** | **72.06** | **67.35** | **64.38** | **62.12** | **60.05** | **58.99** | **55.21** | **+5.64** |

## 5.3 Further Analysis

**Ablation Study**   An ablation study was conducted to investigate the contribution of these components, and Table 3 summarizes their performance on the CIFAR-100 dataset. To preclude interference from performance disparities in the base session with the experimental results, we employ fixed weights derived from the base session and initiate experiments starting from the first new session. The contribution of $\mathcal{R}_{\text{PAMMD}}$ in aligning semantics yields an accuracy improvement of $1.24\%$ compared to the baseline, an advantage further extended to $1.86\%$ by $\mathcal{R}_{\text{VM}}$. At the logits level, $\mathcal{R}_{\text{RC}}$ demonstrates a more pronounced contribution to accuracy enhancement compared to the feature-level rewards, boosting performance from $1.86\%$ to $3.50\%$. This underscores the critical role of feedback from the classifier's decision space in improving the training effectiveness of the generated images. $\mathcal{R}_{\text{CSCA}}$ further highlights the efficacy of generating customized hard samples based on the classifier's mastery of the training data, leading to a substantial accuracy increase to $5.64\%$.

**The Diffusion Model's Intrinsic Baseline**   Using pretrained diffusion models for downstream tasks requires careful consideration of their inherent knowledge's impact on results. To investigate this, we conducted experiments by generating varying quantities of training images, employing only a foundational CLIP-score reward while maintaining consistency across other parameters. Figure 3 illustrates the average classification accuracy in new sessions under different guidance scales. While empirical evidence suggests that a larger volume of generated samples correlates with improved training efficacy, the substantial resource expenditure associated with such large-scale generation is incongruent with the practical constraints of FS-CIL scenarios. Particularly in low-data regimes (*i.e.*, fewer than 50 generated images per class), the performance of baseline methods fails to reach the level achieved by our proposed approach (indicated by the dashed line). DCS offers a approach that enables achieving performance comparable to that obtained with a larger volume of images, even when using a minimal number of generated images under resource-constrained conditions.

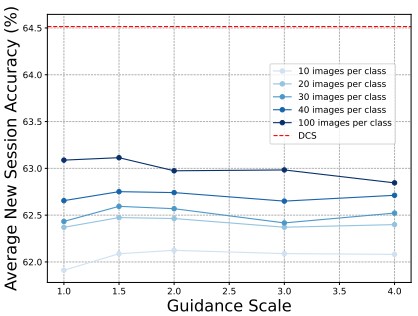

Figure 3: **Training performance comparison.** DCS versus the vanilla diffusion model generating images at varying quantities and guidance scales, with the latter underperforming at comparable generation scales.

## 6   Conclusion

This paper introduced DCS, a novel framework advancing FSCIL by establishing a mutual boosting loop between a diffusion model and a classifier through reward-aligned learning. DCS employs a feature-level as well as logits-level reward system to guide the generation of strategically beneficial images, ensuring they are tailored to the classifier's evolving needs. Empirical validation on benchmark datasets demonstrated DCS's state-of-the-art performance in mitigating catastrophic forgetting and effectively learning new classes from minimal data. The study confirmed that aligning generative processes with classifier goals is crucial for continual learning, emphasizing DCS's role in using diffusion models to address major FSCIL challenges and enhance adaptability.

## Acknowledgments

This work is partially supported by grants from the National Natural Science Foundation of China (No. 62132002, No. 62202010, and No. 62402015), the Beijing Nova Program (No. 20250484786), the China Postdoctoral Science Foundation under grant (No. 2024M750100), the Postdoctoral Fellowship Program of CPSF under grant (No. GZB20230024), and the Fundamental Research Funds for the Central Universities.

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

# A  Overall Pipeline

DCS aims to enhance the generalization ability of the classifier by constructing bidirectional feedback between the diffusion model and the classifier, while only generating a few images. The workflow of DCS is detailed in Algorithm 1. Compared to the original FSCIL learning strategy, DCS only adds the step of generating new images.

In the base session, we first train the classifier using real data. Since the data in this session is sufficient, we focus on generating more challenging samples with higher ambiguity (by adding $\mathcal{R}_{\text{CSCA}}$ based on the feature-level reward) and use them to fine-tune the classifier.

In each incremental session, we first initialize the classifier weights $W_c$ using the limited real data from the new classes (as described in Section 3). Then, we generate additional learning data for both the new and previously learned old classes. For the new classes, to prevent overfitting, DCS aims to help the classifier quickly build an understanding of their concepts. Therefore, we only use $\mathcal{R}_{\text{RC}}$ at the logits level. For the old classes, the focus is on their confusion with the new classes, so $\mathcal{R}_{\text{CSCA}}$ is applied, as in the base session. Finally, we mix the newly generated data for both new and old classes and fine-tune the classifier.

---

**Algorithm 1** FSCIL with DCS

---

1: **Data:** $\mathcal{D}_{base}, \mathcal{D}_{inc}^t$ ($K$-shot, for $t = 1, \ldots, T$)
2: **Models:**
3:      Classifier $\sigma = (f, W)$ with parameters $(\theta_f, \theta_W)$
4:      Diffusion Model $D$
5: **Loss:** $\mathcal{L}_{cls}$
6: **Rewards:** $\mathcal{R}_{\text{PAMMD}}, \mathcal{R}_{\text{VM}}, \mathcal{R}_{\text{RC}}, \mathcal{R}_{\text{CSCA}}$
7: **Generated Images Count:** $N_{base}, N_{new}, N_{old}$ per class

     // Base Session
8: $\mathcal{C}_{base} \leftarrow$ classes in $\mathcal{D}_{base}$
9: $(\theta_f, \theta_W) \leftarrow \arg\min_{\theta_f, \theta_W} \mathcal{L}_{cls}(\sigma(\mathcal{D}_{base}), \mathcal{C}_{base})$
10: $X_{gen\_base} \leftarrow \varnothing$
11: $\mathcal{R}_{base} \leftarrow \mathcal{R}_{\text{PAMMD}} + \mathcal{R}_{\text{VM}} + \mathcal{R}_{\text{CSCA}}$
12: **for all** class $c \in \mathcal{C}_{base}$ **do**                 ▷ Generate images for the base classes
13:      $X_{gen\_base}^c \leftarrow \text{Generate}(D, c, N_{base}, \mathcal{R}_{base}, \sigma)$
14:      $X_{gen\_base} \leftarrow X_{gen\_base} \cup X_{gen\_base}^c$
15: $(\theta_f, \theta_W) \leftarrow \arg\min_{\theta_f, \theta_W} \mathcal{L}_{cls}(\sigma(\mathcal{D}_{base} \cup X_{gen\_base}), \mathcal{C}_{base})$     ▷ Fine-tune classifier
16: $\mathcal{C}_{seen} \leftarrow \mathcal{C}_{base}$

     // Incremental Session
17: **for all** session $t = 1, \ldots, T$ **do**
18:      $\mathcal{C}_{new}^t \leftarrow$ classes in $\mathcal{D}_{inc}^t$
19:      $\mathcal{C}_{old}^t \leftarrow \mathcal{C}_{seen}$
20:      $\theta_W \leftarrow \frac{1}{K} \sum_{i=1}^{K} f(\mathcal{D}_{inc,i}^t)$              ▷ Initialize $\theta_W$ for $\mathcal{C}_{new}^t$ using $\mathcal{D}_{inc}^t$
21:      $X_{gen\_inc} \leftarrow \varnothing$
22:      $\mathcal{R}_{new} \leftarrow \mathcal{R}_{\text{PAMMD}} + \mathcal{R}_{\text{VM}} + \mathcal{R}_{\text{RC}}$
23:      **for all** class $c \in \mathcal{C}_{new}^t$ **do**         ▷ Generate images for the new classes
24:          $X_{gen\_inc}^c \leftarrow \text{Generate}(D, c, N_{new}, \mathcal{R}_{new}, \sigma)$
25:          $X_{gen\_inc} \leftarrow X_{gen\_inc} \cup X_{gen\_inc}^c$
26:      $\mathcal{R}_{old} \leftarrow \mathcal{R}_{\text{PAMMD}} + \mathcal{R}_{\text{VM}} + \mathcal{R}_{\text{CSCA}}$
27:      **for all** class $c \in \mathcal{C}_{old}^t$ **do**          ▷ Generate images for the old classes
28:          $X_{gen\_inc}^c \leftarrow \text{Generate}(D, c, N_{old}, \mathcal{R}_{old}, \sigma)$
29:          $X_{gen\_inc} \leftarrow X_{gen\_inc} \cup X_{gen\_inc}^c$
30:      $\mathcal{D}_{train\_inc}^t \leftarrow \mathcal{D}_{inc}^t \cup X_{gen\_inc}$
31:      $(\theta_f, \theta_W) \leftarrow \arg\min_{\theta_f, \theta_W} \mathcal{L}_{cls}(\sigma(\mathcal{D}_{train\_inc}^t), \mathcal{C}_{new}^t \cup \mathcal{C}_{old}^t)$     ▷ Fine-tune classifier
32:      $\mathcal{C}_{seen} \leftarrow \mathcal{C}_{seen} \cup \mathcal{C}_{new}^t$
33:      Evaluate $\sigma(\cdot; \theta_f, \theta_W)$ on $\mathcal{C}_{seen}$.

---

# B Implementation Details

**Diffusion Model**    As described in Section 5.1, we use Stable Diffusion 3.5 Medium [11] to generate images. For all datasets, the `guidance_scale` is set to 2.0. The prompt is "a photo of {class name}", and the negative prompt is "Anime, smooth, close-up, virtual, logo, partial magnification, exquisite". The number of steps for each image is set to 10.

**Diffusion Alignment as Sampling (DAS) [21]**    For all datasets, the `kl_coeff` is set to 0.001, `num_particles` is set to 16, and `tempering_gamma` is set to 0.008.

**Hyperparameters**    In $\mathcal{R}_{\text{PAMMD}}$, $\alpha = 1$, $\beta = 2$, and $k = 0$ (since this term is constant). In $\mathcal{R}_{\text{RC}}$, $T_{base}$ is set to 2.0, and $T_{scale}$ is set to 1.0. In $\mathcal{R}_{\text{CSCA}}$, $\gamma$ is set to 1.0, and only the top-3 old classes $y_t$ most similar to $y_c$ are considered.

**Training Details**    For the optimizer, SGD is used on all datasets with momentum of 0.9 and weight decay of 0.0005. In addition, we use the cosine annealing strategy to dynamically adjust the learning rate during training. For the CIFAR-100 dataset, the initial learning rate is 0.1 with 50 epochs for the base session, and 0.01 with 5 epochs for the incremental session. For the *mini*ImageNet dataset, the initial learning rate is 0.1 with 120 epochs for the base session, and 0.05 with 30 epochs for the incremental session. For the CUB-200 dataset, the initial learning rate is 0.002 with 120 epochs for the base session, and 0.0005 with 10 epochs for the incremental session. Following [15, 68, 27], we employ ResNet-18 as the backbone for CUB200, and ResNet-12 for *mini*ImageNet and CIFAR100. After the base session, we freeze the shallow layers of ResNet and keep only the last layer for training, with its learning rate individually set to 0.001 times the learning rate of the new class listed above.

**Experimental Environment**    All experiments were performed under Ubuntu 20.04.4 LTS operating system with NVIDIA GeForce RTX 4090 GPU. The experimental code is written in Python 3.8.19, and the PyTorch (version 1.13.0+cu117) is used for the deep learning framework. The source code of the diffusion model used in the experiments is from the open source library `diffusers` [1] (version 0.32.2).

# C More Results

## C.1 Comparison results on the CIFAR-100 Dataset.

In the main paper, we compare different methods in accuracy across all sessions on CUB-200 and *mini*ImageNet. In Table 4, we report the detailed comparison results on CIFAR-100 dataset.

Table 4: Comparison results on CIFAR-100 dataset. * denotes results from [27].

| Methods | Accuracy in each session (%) ↑ | | | | | | | | | Avg ↑ |
|---|---|---|---|---|---|---|---|---|---|---|
| | 0 | 1 | 2 | 3 | 4 | 5 | 6 | 7 | 8 | |
| TOPIC [52] | 64.10 | 55.88 | 47.07 | 45.16 | 40.11 | 36.38 | 33.96 | 31.55 | 29.37 | 42.62 |
| CEC [73] | 73.07 | 68.88 | 65.26 | 61.19 | 58.09 | 55.57 | 53.22 | 51.34 | 49.14 | 59.53 |
| FACT [77] | 74.60 | 72.09 | 67.56 | 63.52 | 61.38 | 58.36 | 56.28 | 54.24 | 52.10 | 62.24 |
| TEEN [60] | 74.92 | 72.65 | 68.74 | 65.01 | 62.01 | 59.29 | 57.90 | 54.76 | 52.64 | 63.10 |
| SAVC [50] | 78.77 | 73.31 | 69.31 | 64.93 | 61.70 | 59.25 | 57.13 | 55.19 | 53.12 | 63.63 |
| DyCR [38] | 75.73 | 73.29 | 68.71 | 64.80 | 62.11 | 59.25 | 56.70 | 54.56 | 52.24 | 63.04 |
| ALFSCIL [26] | 80.75 | **77.88** | **72.94** | **68.79** | **65.33** | **62.15** | 60.02 | 57.68 | 55.17 | **66.75** |
| OrCo* [3] | 79.77 | 63.29 | 62.39 | 60.13 | 58.76 | 56.56 | 55.49 | 54.19 | 51.12 | 60.19 |
| ADBS* [27] | 79.93 | 75.22 | 71.11 | 65.99 | 62.46 | 58.38 | 55.96 | 53.72 | 51.15 | 63.77 |
| DCS(Ours) | **81.09** | 75.96 | 72.06 | 67.35 | 64.38 | 62.13 | **60.05** | **58.99** | **55.21** | 66.36 |

Table 5: Ablation study on generation quality metrics (FID↓, CLIP Score↑). The *vanilla* baseline uses the diffusion model without any reward guidance. The key insight is the *relative improvement* across settings. The degradation in FID/CLIP from adding $R_{\text{CSCA}}$ is by design, as this reward's objective is to generate challenging samples near decision boundaries to improve classifier robustness, not to generate high-fidelity images.

| Reward Combination | FID-CF100↓ | FID-mini↓ | FID-CUB↓ | CLIP-CF100↑ | CLIP-mini↑ | CLIP-CUB↑ |
|---|---|---|---|---|---|---|
| w/o reward (vanilla) | 142.6 | 135.8 | 152.1 | 68.3 | 69.5 | 67.1 |
| $R_{\text{PAMMD}}$ | 128.7 | 120.4 | 139.3 | 72.3 | 73.4 | 70.6 |
| $+ R_{\text{VM}}$ | **122.6** | **113.8** | **133.1** | 74.8 | 76.3 | 72.5 |
| $+ R_{\text{VM}} + R_{\text{RC}}$ | 123.4 | 114.3 | 134.3 | **78.8** | **80.2** | **76.1** |
| $+ R_{\text{VM}} + R_{\text{CSCA}}$ | 133.0 | 124.2 | 144.4 | 71.9 | 72.0 | 70.2 |

## C.2  Extended Ablation Studies

To complement the ablation study in the main paper, we provide further analyses on the impact of our reward components on generation quality, the necessity of each component, and the generalizability of the reward design across different datasets.

### C.2.1  Ablation on Generation Quality Metrics

In addition to downstream task performance, an analysis was conducted on the impact of our reward components on standard generation quality metrics, namely the Fréchet Inception Distance (FID) and CLIP Score. Under the data-scarce conditions of FSCIL, the estimation of such metrics from a limited number of samples (e.g., 5-30) is subject to high variance and may not fully represent the true distribution quality. Accordingly, the following evaluation prioritizes the **relative improvement** across reward configurations over absolute scores, as the latter are not directly comparable to benchmarks in large-scale generation literature. Emphasizing relative gains provides a more precise assessment of the targeted effect of each reward component.

The ablation study was performed on the first class of the first incremental session for CIFAR-100, *mini*ImageNet, and CUB-200. For each experimental setting, 30 images were generated, and the average scores are reported in Table 5. The results indicate that the feature-level rewards, $R_{\text{PAMMD}}$ and $R_{\text{VM}}$, significantly improve both FID and CLIP scores by promoting semantic consistency and aligning the feature distribution with real data. The inclusion of $R_{\text{RC}}$ leads to the best CLIP scores, as it directly optimizes for classification confidence, pushing generated samples to be more semantically pure. Conversely, adding the confusion-aware reward $R_{\text{CSCA}}$ degrades these metrics. This result validates our hypothesis: the goal of $R_{\text{CSCA}}$ is to generate strategically challenging samples at the decision boundary, which are by nature less aligned with the clean data distribution. This demonstrates that our reward system can generate not only high-fidelity images but also targeted samples tailored to the classifier's evolving needs.

### C.2.2  Ablation on Reward Component Necessity

To demonstrate that all four reward components are necessary for the final performance, we conducted a *leave-one-out* analysis on the CIFAR-100 dataset. As shown in Table 6, removing any single component from the full DCS framework results in a performance decrease, confirming that all components contribute synergistically.

Table 6: Leave-one-out ablation study on CIFAR-100. Removing any component degrades the final average accuracy, demonstrating their synergistic contribution.

| Method | Avg. Acc. (%) |
|---|---|
| **DCS (Full Model)** | **66.36** |
| DCS w/o $R_{\text{PAMMD}}$ | 65.12 |
| DCS w/o $R_{\text{VM}}$ | 65.54 |
| DCS w/o $R_{\text{RC}}$ | 64.07 |
| DCS w/o $R_{\text{CSCA}}$ | 64.68 |

Table 7: Sequential ablation studies on *mini*ImageNet and CUB-200, showing consistent performance improvement as reward components are added.

| *mini*ImageNet | | CUB-200 | |
|---|---|---|---|
| **Method** | **Avg. Acc. (%)** | **Method** | **Avg. Acc. (%)** |
| Baseline (w/o reward) | 64.03 | Baseline (w/o reward) | 65.21 |
| + $R_{\text{PAMMD}}$ | 65.28 | + $R_{\text{PAMMD}}$ | 66.35 |
| + $R_{\text{PAMMD}}$ + $R_{\text{VM}}$ | 65.57 | + $R_{\text{PAMMD}}$ + $R_{\text{VM}}$ | 66.89 |
| + $R_{\text{PAMMD}}$ + $R_{\text{VM}}$ + $R_{\text{RC}}$ | 66.85 | + $R_{\text{PAMMD}}$ + $R_{\text{VM}}$ + $R_{\text{RC}}$ | 67.41 |
| **DCS (Full Model)** | **68.14** | **DCS (Full Model)** | **69.73** |

### C.2.3 Generalizability of the Reward Design

To confirm that the effectiveness of our reward design is not confined to a single dataset, we performed the same sequential *add-one* ablation study on the *mini*ImageNet and CUB-200 datasets. The results, presented in Table 7, show a consistent and positive trend in performance gains as each reward component is added, demonstrating the general applicability of our approach across diverse benchmarks.

## C.3 Qualitative Analysis of Generated Images

Figure 4 displays a qualitative comparison, where the first column shows original images from the *mini*ImageNet dataset and the remaining columns show examples generated by our method. Guided by text prompts and our multi-faceted reward system, the generated images are not only semantically correct but are also rich in scenic diversity.

## D  Incremental Computation of Reward Functions

This section details the incremental update rules for the Prototype-Anchored Maximum Mean Discrepancy Reward ($\mathcal{R}_{\text{PAMMD}}$) and the Dimension-Wise Variance Matching Reward ($\mathcal{R}_{\text{VM}}$). These rules allow for efficient computation as the diffusion model generates images sequentially. We denote the feature vector of the $k$-th generated image as $\boldsymbol{z}_k$.

### D.1  Incremental Computation of $\mathcal{R}_{\text{PAMMD}}$

Recall the $\mathcal{R}_{\text{PAMMD}}$ formula for a set of $N$ generated features $\mathcal{I}_{\text{gen}}^{(N)}$ and a class prototype $\boldsymbol{\mu}_c$:

$$\mathcal{R}_{\text{PAMMD}}(\boldsymbol{x}_{\text{gen}}, \mathcal{I}_{\text{gen}}^{(c,N)}) = -\frac{\alpha}{N^2} \underbrace{\sum_{i=1}^{N}\sum_{j=1}^{N} k(\boldsymbol{z}_i, \boldsymbol{z}_j)}_{\text{Diversity}} + \frac{\beta}{N} \underbrace{\sum_{i=1}^{N} k(\boldsymbol{z}_i, \boldsymbol{\mu}_c)}_{\text{Consistency}}, \tag{16}$$

We need to incrementally update term *Diversity* and term *Consistency* when a new $(N)$-th feature $\boldsymbol{z}_N$ is generated and added to a set of $N-1$ existing features.

Let

$$S_D^{(N-1)} = \text{Diversity}_{N-1} = \sum_{i=1}^{N-1}\sum_{j=1}^{N-1} k(\boldsymbol{z}_i, \boldsymbol{z}_j), \tag{17}$$

$$S_C^{(N-1)} = \text{Consistency}_{N-1} = \sum_{i=1}^{N-1} k(\boldsymbol{z}_i, \boldsymbol{\mu}_c). \tag{18}$$

**(1) Initialization ($N = 0$)**

Before any image is generated for the class:

$$S_D^{(0)} = 0, \tag{19}$$

$$S_C^{(0)} = 0. \tag{20}$$

**(2) First Sample ($N = 1$)**

When the first image feature $\boldsymbol{z}_1$ is generated:

$$S_D^{(1)} = k(\boldsymbol{z}_1, \boldsymbol{z}_1), \tag{21}$$

$$S_C^{(1)} = k(\boldsymbol{z}_1, \boldsymbol{\mu}_c), \tag{22}$$

**(3) Subsequent Samples ($N > 1$)**

Assume we have $S_D^{(N-1)}$ and $S_C^{(N-1)}$ for $N-1$ samples, and a new feature $\boldsymbol{z}_N$ is generated. The new sum $S_D^{(N)}$ includes the old sum $S_D^{(N-1)}$, the self-similarity of the new sample $k(\boldsymbol{z}_N, \boldsymbol{z}_N)$, and twice the sum of similarities between the new sample and all $N-1$ previous samples (due to symmetry $k(\boldsymbol{z}_N, \boldsymbol{z}_i) + k(\boldsymbol{z}_i, \boldsymbol{z}_N)$ where $k$ is symmetric:

$$S_D^{(N)} = S_D^{(N-1)} + k(\boldsymbol{z}_N, \boldsymbol{z}_N) + 2 \sum_{i=1}^{N-1} k(\boldsymbol{z}_N, \boldsymbol{z}_i). \tag{23}$$

The new sum $S_C^{(N)}$ is the old sum $S_C^{(N-1)}$ plus the similarity of the new sample to the prototype:

$$S_C^{(N)} = S_C^{(N-1)} + k(\boldsymbol{z}_N, \boldsymbol{\mu}_c). \tag{24}$$

With $S_D^{(N)}$ and $S_C^{(N)}$, the $\mathcal{R}_{\text{PAMMD}}(\mathcal{I}_{\text{gen}}^{(N)}, \boldsymbol{\mu}_c)$ can be calculated using the Equation (16). To compute $\sum_{i=1}^{N-1} k(\boldsymbol{z}_N, \boldsymbol{z}_i)$, we need access to the feature vectors of the $N-1$ previously accepted images.

### D.2 Incremental Computation of $\mathcal{R}_{\text{VM}}$

Recall the $\mathcal{R}_{\text{VM}}$ for a candidate $\boldsymbol{z}_N$ when added to a set of $N-1$ generated features, forming an augmented set $\mathcal{I}_{\text{gen}}^{(N)}$:

$$\mathcal{R}_{\text{VM}}(\boldsymbol{x}_{\text{gen}}, \mathcal{I}_{\text{gen}}^{(c,N)}) = -\sum_{d=1}^{D} \left( v_{\text{gen},[d]} - v_{\text{real},[d]} \right)^2, \tag{25}$$

where $v_{\text{gen},[d]} = \text{Var}(\{f(\boldsymbol{x}_j)^d \mid \boldsymbol{x}_j \in \mathcal{I}_{\text{gen}}\})$ is the sample variance of the $d$-th dimension of features in $\mathcal{I}_{\text{gen}}$, and $v_{\text{real},[d]}$ is defined analogously.

The sample variance for a set of $N$ values $\{\boldsymbol{z}_1, \ldots, \boldsymbol{z}_N\}$ is given by:

$$\text{Var}(\{\boldsymbol{z}_i\}) = \frac{1}{N-1} \sum_{i=1}^{N} (\boldsymbol{z}_i - \bar{\boldsymbol{z}})^2 = \frac{1}{N-1} \left( \sum_{i=1}^{N} \boldsymbol{z}_i^2 - N\bar{\boldsymbol{z}}^2 \right) = \frac{1}{N-1} \left( \sum_{i=1}^{N} \boldsymbol{z}_i^2 - \frac{1}{N} \left( \sum_{i=1}^{N} \boldsymbol{z}_i \right)^2 \right). \tag{26}$$

For $N = 0$ and $N = 1$, variance is typically undefined or taken as 0. We consider $N \geq 2$ for variance calculation. To compute $v_{\text{gen},[d]}$ incrementally for dimension $d$, when a new feature $\boldsymbol{z}_N$ (with $d$-th component $z_{N,[d]}$) is added, we need to maintain the sum of values and the sum of squared values for each dimension $d$ of the generated features.

For each dimension $d \in \{1, \ldots, D\}$, the sum of feature values in dimension $d$ for $N-1$ samples is:

$$\text{Sum}_d^{(N-1)} = \sum_{i=1}^{N-1} z_{i,[d]}. \tag{27}$$

The sum of squared feature values in dimension $d$ for $N-1$ samples is:

$$\text{SumSq}_d^{(N-1)} = \sum_{i=1}^{N-1} (z_{i,[d]})^2. \tag{28}$$

**(1) Initialization ($N = 0$)**

Before any image is generated, for each dimension $d$:

$$\text{Sum}_d^{(0)} = 0, \tag{29}$$

$$\text{SumSq}_d^{(0)} = 0, \tag{30}$$

$$v_{\text{gen},[d]} = 0. \tag{31}$$

**(2) First Sample ($N = 1$)**

When the first feature $\boldsymbol{z}_1$ is generated:

$$\text{Sum}_d^{(1)} = z_{1,[d]}, \tag{32}$$

$$\text{SumSq}_d^{(1)} = (z_{1,[d]})^2, \tag{33}$$

$$v_{\text{gen},[d]} = 0. \tag{34}$$

**(3) Subsequent Samples ($N > 1$)**

Assume we have $\text{Sum}_d^{(N-1)}$ and $\text{SumSq}_d^{(N-1)}$ for $N-1$ samples, and a new feature $\boldsymbol{z}_N$ (with $d$-th component $z_{N,[d]}$) is generated:

$$\text{Sum}_d^{(N)} = \text{Sum}_d^{(N-1)} + z_{N,[d]}, \tag{35}$$

$$\text{SumSq}_d^{(N)} = \text{SumSq}_d^{(N-1)} + (z_{N,[d]})^2, \tag{36}$$

$$v_{\text{gen},[d]} = \frac{1}{N-1} \left( \text{SumSq}_d^{(N)} - \frac{1}{N}(\text{Sum}_d^{(N)})^2 \right). \tag{37}$$

With $v_{\text{gen},[d]}$ calculated for all dimensions $d$ using the updated sums, the $\mathcal{R}_{\text{VM}}$ can be computed.

# E    Discussion

## E.1    Limitations

The proposed framework's performance is contingent upon access to high-quality, pre-trained diffusion models. Furthermore, the efficacy of DCS may be reduced when applied to highly specialized domains poorly represented in the diffusion model's training data, and the framework's multi-component reward system and iterative boosting loop introduce complexities in tuning and computational demand.

## E.2    Broader impact

The DCS framework is a foundational research contribution aimed at improving the adaptability of machine learning models in data-scarce, continual learning settings. As the approach leverages a pre-trained diffusion model, the nature and characteristics of the generated images are directly determined by this underlying component. Consequently, the framework inherits any potential societal impacts, such as fairness considerations or biases, from the foundational model it employs. The safety and security of the generated content are therefore contingent on the specific diffusion model used, and we encourage practitioners to consider the ethical implications of the foundational model selected for any given application.

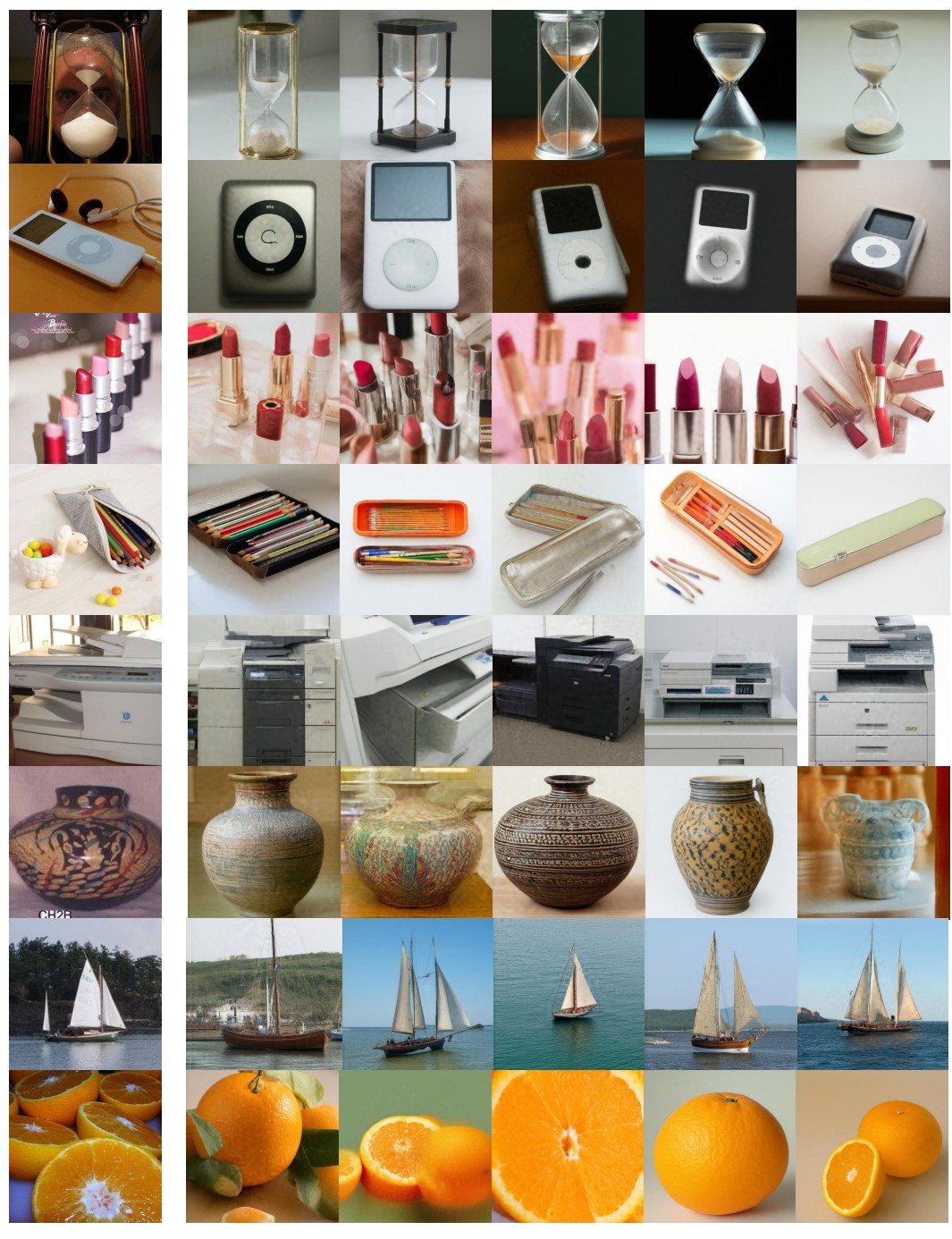

Figure 4: Qualitative comparison of real and generated images on *mini*ImageNet. The first column displays real images from the dataset. The subsequent columns show diverse and semantically correct images generated by our DCS framework for the corresponding classes.

