# OpenReview forum: "Diffusion-Classifier Synergy: Reward-Aligned Learning via Mutual Boosting Loop for FSCIL"
_NeurIPS.cc/2025/Conference — NeurIPS 2025 poster_

### Official Review · Reviewer_doaJ · 2025-06-25

**Clarity:** 3
**Significance:** 3
**Originality:** 2
**Rating:** 4
**Confidence:** 4

**Summary:**

This paper introduces Diffusion-Classifier Synergy (DCS) that tackles data scarcity and the stability-plasticity dilemma for FSCIL.  DCS establishes a co-evolutionary, mutual boosting loop between a diffusion model and an FSCIL classifier. The core of the framework is a reward-aligned learning strategy where a multi-faceted reward function guides the diffusion model to generate strategically beneficial images. This reward system operates at feature and logit levels. By having the improved classifier provide more refined reward signals back to the diffusion model, DCS achieves state-of-the-art performance on FSCIL benchmarks.

**Questions:**

Please see the weaknesses.

**Ethical Concerns:**

["NO or VERY MINOR ethics concerns only"]

**Final Justification:**

The authors have adequately addressed my concerns. Therefore, I decided to maintain the score.

**Limitations:**

Yes, in the supplementary material.

**Quality:**

3

**Strengths And Weaknesses:**

Strength:

- The concept of a "mutual boosting loop" is departure from conventional methods that use generative models as a one-way data provider.
- The paper designs a multi-faceted reward function that operates at both the feature and logits levels.
- The framework is designed to solve the key challenges of using diffusion models in FSCIL: semantic misalignment, lack of diversity, and inefficient, non-adaptive feedback.  The ablation study demonstrates that each proposed reward component contributes positively to the final performance.


Weaknesses:

- The method's superiority is not consistent across all datasets presented. For example, while it shows a good result on miniImageNet, its average accuracy on CIFAR-100 is slightly below that of a previously published method, ALFSCIL (66.75%). Also, the improvement on CUB is marginal. The paper does not analyze or explain this discrepancy between datasets.

- he framework introduces significant computational overhead compared to traditional FSCIL methods. It requires integrating a large, pre-trained diffusion model (Stable Diffusion 3.5 Medium) and running an iterative, reward-based sampling algorithm (DAS).  The paper lacks a direct comparison of training time or computational cost against other FSCIL baselines, which is a critical factor for evaluating practical applicability.

- The experiments use generated images at a 512x512 resolution, which are then resized.  This is a major deviation from standard FSCIL protocols, where datasets like CIFAR-100 (32x32) and miniImageNet (84x84) are used at their native, much smaller resolutions. This use of a high-resolution generator could provide an advantage not available to compared methods and affects the fairness of the comparison.

- While the paper is built on generating "strategically beneficial images," it does not provide any visual examples of these images or analysis of how the generated images differ from the real images.

---

> ### Author Rebuttal · Authors · 2025-07-29
>
> We sincerely thank you for your careful review and encouraging recognition of our work. We will respond to the questions you have raised in detail below.
>
>
>
> > **Q1: Inconsistent performance superiority across all datasets.**
>
> **A1:** Thank you for this insightful comment. We appreciate the opportunity to provide a more nuanced analysis of our model's performance.
>
> We agree that a holistic evaluation across multiple metrics is crucial for FSCIL. While average accuracy provides a general overview, we argue that **final session accuracy** and forgetting rate are more critical metrics for evaluating true continual learning, as they measure a model's long-term viability.
>
> On CIFAR-100, our method demonstrates superior knowledge retention. From session 1 to 8, ALFSCIL's accuracy drops by 22.71% (from 77.88% to 55.17%), whereas DCS's accuracy drops by only 20.75% (from 75.96% to 55.21%). Crucially, our performance in the final session is higher than ALFSCIL's, highlighting our model's enhanced ability to both learn new concepts and mitigate catastrophic forgetting.
>
> Furthermore, on the miniImageNet benchmark, DCS significantly outperforms ALFSCIL across the board: our average accuracy is 3.34% higher (68.14% vs. 64.80%), and our final session accuracy is 4.68% higher (57.99% vs. 53.31%). On the fine-grained CUB-200 dataset, while SAVC's average accuracy (69.35%) is the closest to ours (69.73%), DCS achieves a higher final session accuracy (63.40% vs. SAVC's 62.50%) and shows a smaller accuracy drop from the first to the last incremental session, indicating better long-term learning.
>
> **The performance variation stems from dataset characteristics.** CUB-200's fine-grained nature presents a unique challenge due to high inter-class similarity. We see it as a strength that DCS, with a unified framework and without dataset-specific hyperparameter tuning, achieves consistent and significant improvements on diverse datasets. This demonstrates the generalizability of our core concept. However, we agree that exploring reward designs tailored to the semantic distributions of fine-grained datasets is a valuable direction for future research.
>
> We hope this clarifies the significant advantages of our approach. Thank you again for your valuable feedback.
>
>
>
> > **Q2: Concerns on Computational Efficiency.**
>
> **A2:** The primary computational overhead of our method is concentrated in the image generation phase that precedes the training of each incremental session. It is important to note that our approach, by leveraging the training-free DAS algorithm, does not require any fine-tuning of the diffusion model itself.
>
> In our experiments, the highly parallelizable image generation for a typical incremental session takes **5-10 minutes** (30 images for each new class and 5-10 for each old class). The subsequent classifier training on this augmented data is very fast, taking **less than a minute** per session. Given the limited number of sessions in standard FSCIL benchmarks, we believe this time overhead is a reasonable and acceptable trade-off for the performance gains achieved.
>
> Crucially, our method adds **no overhead at inference time**. The diffusion model is discarded after training, and the final classifier is a standard ResNet. Therefore, its architecture, parameter count, and inference speed are **identical to the baseline**.
>
> Finally, the computational cost is flexible and can be further optimized by generating fewer samples, adjusting the DAS configuration, or employing a more lightweight diffusion model, allowing users to balance performance with available resources. We will add a detailed analysis of this in the final version of the paper.
>
>
>
>
>
>
>
> > **Q3: Concerns regarding the fairness of comparison due to the use of high-resolution image generation.**
>
> **A3:** We would like to respectfully clarify a potential misunderstanding. While the images were initially generated at a $512 \times 512$ resolution, they were **always resized down to the native resolution of the respective dataset** (e.g., $32 \times 32$ for CIFAR-100) using `torchvision.transforms.Resize` **before** being input to the classifier.
>
> Crucially, this downscaling process is inherently lossy; the resulting low-resolution image does not contain more information than an original image of the same size and is simply a synthetic sample that conforms to the standard data format. Therefore, the model was not trained on high-resolution images in any form, **ensuring no unfair informational advantage** was conferred.
>
> The decision to generate at a higher resolution was a practical necessity. The performance of the Stable Diffusion model we employed is optimized for specific resolutions. This is because its core model was pre-trained on images of larger dimensions, such as $512 \times 512$, $1024 \times 1024$, or even higher. At the very low native resolutions of the FSCIL datasets, the model is **unable to generate** coherent, semantically meaningful images, often producing only noise. Our approach allows us to use this state-of-the-art generative tool effectively, while still maintaining a fair and standardized evaluation protocol for the classifier.
>
>
>
>
>
> > **Q4: Lack of visual examples and analysis for the images.**
>
> **A4:** We agree that visualizing the generated images is crucial for a complete understanding of our method's mechanics. We will certainly incorporate a dedicated section with visual examples and a comparative analysis in the final version of the paper.
>
> Due to the format limitations of this rebuttal, we are unable to display images directly. Instead, we offer the following textual description of their key characteristics:
>
> *   **Semantic Correctness and Style Diversity:** Guided by both text prompts and our multi-faceted reward system, the generated images are not only semantically correct but also rich in scenic diversity. Since our diffusion model is guided by the classifier’s state rather than the original dataset's images, the generated samples exhibit a broader range of styles than the source data. This process, constrained by our logits-level rewards, ensures the classifier can correctly identify these novel variations, thereby significantly enhancing its generalization.
>
> *   **Subject Amplification:** A key difference from real images lies in the portrayal of the subject. Our method provides a clear semantic target, prompting the diffusion model to amplify the subject within the frame, often resulting in a larger area and finer detail. While this helps the classifier focus on and learn richer class-specific features, it does differ from many real-world images where subjects can be small or part of a complex, multi-object scene. Although the content of the generated image can be constrained by designing richer text prompts, we believe that developing more adaptive algorithms to generate scenes more aligned with the original dataset is a promising direction for research.
>
> *   **Nature of "Hard" Samples:** For images generated to be intentionally confusing (e.g., an 'X' that fools the classifier into seeing a 'Y' via the `R_CSCA` reward), the results are particularly interesting. Visually, the generated image remains clearly identifiable as 'X' to a human observer and does not appear as a blend of 'X' and 'Y'. However, these images contain subtle, high-frequency patterns, akin to adversarial noise. As our reward signals confirm, these patterns are highly effective at misleading the classifier. We hypothesize this is because the diffusion model learns to exploit the classifier's high sensitivity to texture and fine details, which humans often overlook in favor of holistic shapes. A thorough investigation of this adversarial-like phenomenon is a compelling area for future work.

---

> > ### Comment · Reviewer_doaJ · 2025-08-02
> >
> > Thank you for your comprehensive and thoughtful response to my review. I appreciate the detailed clarifications as well as the additional analysis you have provided. They are a great addition to the paper.

---

> > > ### Author Response · Authors · 2025-08-02
> > >
> > > Thank you for the encouraging feedback. We're very pleased that our response and the new analysis have effectively addressed your points. We sincerely appreciate your time and constructive comments throughout the review process.

---

### Official Review · Reviewer_eG2t · 2025-06-30

**Clarity:** 3
**Significance:** 3
**Originality:** 3
**Rating:** 4
**Confidence:** 4

**Summary:**

This paper presents a reward-alignment-based method for Few-Shot Class-Incremental Learning (FSCIL).
Different from previous works that operate on representative spaces or use meta-learning, this work suggests using a diffusion model to generate data for few-shot learning.
To alleviate the fidelity-diversity challenge and Inefficient feedback from the classifier, the paper presents a two-level (i.e., feature- and logits-level) reward system.
The authors show experiments for the effectiveness of the proposed methods.

**Questions:**

Please see the weakness above.

**Ethical Concerns:**

["NO or VERY MINOR ethics concerns only"]

**Final Justification:**

Thanks for the rebuttal provided by the authors. I think most of my concerns are addressed, so I will keep my original positive score unchanged.

**Limitations:**

Yes

**Quality:**

3

**Strengths And Weaknesses:**

Strengths:
1. New insights might be provided by the generative-based methods for FSCIL.
2. The design of rewards is with strong motivation.
3. The experiment results shown are with convincing improvement.

Weaknesses:
1. The first concern is the technical contribution of this work. It is great to see the incorporation of rewards into diffusion models during FSCIL. However, the proposed method is heavily based on Diffusion Alignment as Sampling (DAS).
2. Lack of training time analysis during training, since the proposed method needs the diffusion model for generating images, it should be clarified if this significantly extends the training process.
3. I think it lacks detailed comparisons of the proposed method and previous generative-based incremental learning methods like [1-2].

Ref:

[1]Gao, Rui, and Weiwei Liu. "Ddgr: Continual learning with deep diffusion-based generative replay." International Conference on Machine Learning. PMLR, 2023.

[2]Meng, Zichong, et al. "Diffclass: Diffusion-based class incremental learning." European Conference on Computer Vision. Cham: Springer Nature Switzerland, 2024.

---

> ### Author Rebuttal · Authors · 2025-07-29
>
> We sincerely thank you for your careful review and encouraging recognition of our work. We will respond to the questions you have raised in detail below.
>
>
>
> > **Q1: Clarity on Technical Contribution Beyond DAS.**
>
> **A1:** Thank you for your positive feedback. This is an insightful question, and we appreciate the opportunity to clarify the core novelty of our work.
>
> We agree that our method is built upon an alignment algorithm, for which we chose Diffusion Alignment as Sampling (DAS). As we state in the Preliminary section, having an effective, training-free method to align the diffusion model with a reward signal is a foundational premise for our framework. Indeed, our approach is not strictly limited to DAS. Any similar technique that achieves this goal, such as those proposed in [a] or [b], could be integrated into our system.
>
> The central technical contribution of our work, Diffusion-Classifier Synergy (DCS), is the design of the **mutual boosting loop** itself. Our focus is on how to leverage the generated data distribution to create a controllable, joint co-evolutionary framework for a downstream task. The classifier's evolving state provides a rich, multi-faceted reward signal that guides the diffusion model to generate strategically beneficial samples, which in turn are used to refine the classifier's understanding. This concept of co-evolution is particularly crucial in extreme scenarios like FSCIL, where data is scarce and every new sample must be maximally informative.
>
> In the future, we will certainly explore using novel distribution metrics and target alignment algorithms to optimize the sampling step. Thank you again for your valuable suggestion, which points toward an important direction for the future improvement of our work.
>
> **Ref:**
>
> [a] Cheng, Min et al. "Diffusion Blend: Inference-Time Multi-Preference Alignment for Diffusion Models."
>
> [b] Li, Xiner et al. "Dynamic Search for Inference-Time Alignment in Diffusion Models."
>
>
> > **Q2: Lack of analysis of training time.**
>
> **A2:** The primary computational overhead of our method is concentrated in the image generation phase that precedes the training of each incremental session. It is important to note that our approach, by leveraging the training-free DAS algorithm, does not require any fine-tuning of the diffusion model itself.
>
> In our experiments, the highly parallelizable image generation for a typical incremental session takes **5-10 minutes** (30 images for each new class and 5-10 for each old class). The subsequent classifier training on this augmented data is very fast, taking **less than a minute** per session. Given the limited number of sessions in standard FSCIL benchmarks, we believe this time overhead is a reasonable and acceptable trade-off for the performance gains achieved.
>
> Crucially, our method adds **no overhead at inference time**. The diffusion model is discarded after training, and the final classifier is a standard ResNet. Therefore, its architecture, parameter count, and inference speed are **identical to the baseline**.
>
> Finally, the computational cost is flexible and can be further optimized by generating fewer samples, adjusting the DAS configuration, or employing a more lightweight diffusion model, allowing users to balance performance with available resources. We will add a detailed analysis of this in the final version of the paper.
>
>
>
>
> > **Q3: A detailed comparison between the proposed method and previous generative-based incremental learning methods.**
>
> **A3:** Thank you for your insightful suggestion. We agree that a detailed comparison with related generative-based methods would strengthen our paper, and we will incorporate this discussion into our final version. We hope this clarifies the distinctions and highlights the specific contributions of DCS in the challenging FSCIL context.
>
> We would first like to clarify a key distinction in the problem settings. The cited works, DDGR and DiffClass, primarily address the **general Class-Incremental Learning (CIL)** problem. In CIL, while the data for new tasks may be limited, it is relatively sufficient, and the main goal is to mitigate catastrophic forgetting. Our work focuses on the more challenging **Few-Shot Class-Incremental Learning (FSCIL)** setting, where new classes are introduced with extremely few samples (e.g., 5-shot). This not only exacerbates forgetting but also makes it incredibly difficult for the model to learn robust new concepts.
>
> Due to this fundamental difference in task settings and evaluation protocols, a direct comparison of experimental results is not feasible. However, a comparison of the design philosophies is indeed valuable and reveals the unique contributions of our DCS framework.
>
> ---
>
> ##### **Comparison with DDGR**
>
> *   **Interaction Mechanism**
>     *   **DDGR** employs a direct, gradient-based intervention. It calculates the classifier loss's gradient with respect to the noisy image at each sampling step and injects this gradient to steer the generation. This is a form of micro-level, direct manipulation.
>     *   **DCS** utilizes a more flexible Reward-Aligned Learning paradigm. We design a multi-faceted reward function that assesses the strategic value of a generated sample based on the classifier's state. Then, we use the DAS algorithm to optimize the sampling process to maximize this reward. This is a macro-level, goal-oriented guidance.
> *   **Guidance Signal Complexity**
>     This is the core difference and our key innovation.
>     *   **DDGR's** guidance signal has a singular objective: improving the classification probability of the generated sample.
>     *   **DCS** designs a sophisticated, two-level reward system that goes beyond simple accuracy.
> *   **Strategic Goal**
>     *   **DDGR** primarily aims for high-quality replay of past data for rehearsal.
>     *   **DCS** has a broader strategic goal. It not only replays old data but also performs crucial data augmentation for the new few-shot classes and, as mentioned, generates targeted hard samples to enhance the classifier's discriminative power. It provides data for "review," "preview," and "remedial lessons."
>
> ---
>
> ##### **Comparison with DiffClass**
>
> * **Core Paradigm**
>
>   *   **DiffClass** adopts a "preventative alignment" approach. At the start of each incremental step, it fine-tunes the diffusion model to align the generated data distribution with real and historical data. **The information flow is one-way**: from the generator to the classifier.
>   *   **DCS** establishes a "co-evolutionary synergy." It creates a continuous, bidirectional feedback loop where the classifier's real-time state dynamically guides the diffusion model's generation, leading to a synergistic improvement of both models.
>
> * **Generation Mechanism**
>
>   This represents a fundamental distinction in the technical implementation of the two methods.
>
>   *   **DiffClass**: Employs a **model fine-tuning** approach. At each incremental stage, it leverages LoRA and theMulti-Distribution Matching loss function to actively train and modify the weights of the diffusion model. This process involves gradient computation and model parameter updates.
>   *   **DCS**: Adopts a **training-free guidance** methodology.  Throughout the entire generation process, the weights of the diffusion model remain fixed and unchanged.
>
> * **Guidance Signal Complexity**
>
>   *   **DiffClass** relies on a Multi-Distribution Matching loss, which is a macro-level control that aligns the overall statistics of data distributions.
>   *   **DCS's** reward system offers precision guidance. It operates at a granular level, considering not just distribution alignment but the strategic utility of each sample.

---

> > ### Comment · Reviewer_eG2t · 2025-08-06
> >
> > Thanks for the rebuttal provided by the authors.
> > I think most of my concerns are addressed, so I will keep my original positive score unchanged.

---

> > > ### Author Response · Authors · 2025-08-06
> > >
> > > Thank you for your positive response. We appreciate your constructive feedback throughout this process and will be sure to incorporate your suggestions into the final version of the manuscript to strengthen the work.

---

> ### Comment · Area_Chair_grJe · 2025-08-06
> **Reminder**
>
> Dear Reviewer eG2t,
>
> This is a friendly reminder to check the authors' rebuttal and adjust your rating if necessary. Thanks for your contributions to the NeurIPS reviewing process.
>
> Thanks,
>
> Your AC

---

### Official Review · Reviewer_Dm5X · 2025-07-01

**Clarity:** 4
**Significance:** 3
**Originality:** 2
**Rating:** 4
**Confidence:** 2

**Summary:**

This paper proposes Diffusion-Classifier Synergy (DCS), a reward-aligned framework for Few-Shot Class-Incremental Learning (FSCIL). DCS establishes a sampling loop between a pre-trained diffusion model and a frozen classifier, where the diffusion model generates class-conditional samples guided by classifier-derived rewards. The reward system comprises four components operating at feature and logits levels: prototype-anchored MMD (RPAMMD), dimension-wise variance matching (RVM), recalibrated confidence (RRC), and cross-session confusion-aware rewards (RCSCA). These rewards are integrated into a diffusion sampling framework (based on DAS) to produce informative, diverse, and semantically aligned images. The method achieves strong performance on CIFAR-100, miniImageNet, and CUB-200 benchmarks, using a small number of synthetic samples per class.

**Questions:**

1.	To properly isolate the contribution of your framework, can you provide a comparison against a baseline that also uses Stable Diffusion 3.5 but employs a simpler generation strategy (e.g., vanilla data augmentation without reward guidance, or a simpler guidance mechanism)?
2.	Have you tested the reward mechanism on alternative generative models (e.g., GANs or VAEs)? Would the same guidance strategy work in a non-diffusion setting?
3.	Have you investigated the potential risks of the RCSCA strategy, such as polluting the inter-class boundary and leading to poorer generalization? Do you have any mechanisms to prevent or monitor such negative effects?
4.	Is it feasible to apply DCS with a different encoder? How sensitive is the reward feedback to the encoder quality?

**Ethical Concerns:**

["NO or VERY MINOR ethics concerns only"]

**Final Justification:**

The authors address my concern well during the rebuttal phase. Therefore, I give a positive rating.

**Limitations:**

yes

**Quality:**

3

**Strengths And Weaknesses:**

Strengths：
1.	The paper accurately identifies two key challenges in using generative augmentation for FSCIL: semantic misalignment of generated samples and their ineffective utility for classifier training , proposing a structured, reward-based solution that addresses both.
2.	The framework is training-free on the generator side and operates under realistic resource constraints (generating a small number of samples per class) , which makes it practically appealing.
3.	The paper is exceptionally well-written. The motivation is strong, the methodology is well-structured, and the diagrams (e.g., Figure 1) are highly illustrative.

Weaknesses:
1.	The authors assert that “DCS achieves the highest accuracy in each session”. However, in the comparison on the CIFAR-100 dataset (Table 1), the accuracy of DCS is partly and averagely lower than that of the compared methods ALFSCIL. The more objective academic practice is to highlight the best-performing result in each column, regardless of which method achieved it.
2.	The DCS framework's construction relies on an intricate combination of four different reward functions and a guided sampling algorithm (DAS). The authors do not sufficiently justify why this specific combination of four rewards is necessary, nor do they discuss potential redundancies or conflicts between them. For instance, how is the optimization balanced when RCSCA aims to generate confusing samples while other rewards aim for semantic consistency?
3.	The analysis of the reward components is confined almost entirely to the CIFAR-100 dataset . This lack of testing on other benchmarks like miniImageNet or CUB-200 means the generality of the reward design is unverified.
4.	The methodology of the ablation study in Table 3 is sequential “add-one” approach, which fails to prove that all four components are necessary for the final model's performance. The “leave-one-out” analysis(each component is individually removed from the full four-reward system) is required to establish true necessity. It is plausible that a simpler combination of rewards (e.g., only the two powerful logits-level rewards, RRC and RCSCA) might offer a better or more efficient performance-to-complexity trade-off.
5.	No comparisons are provided against other generative frameworks (e.g., GANs or VAEs), nor against non-reward-guided diffusion generation. This weakens the attribution of gains to the proposed strategy.

---

> ### Author Rebuttal · Authors · 2025-07-29
>
> We sincerely thank you for your careful review and encouraging recognition of our work. We will respond to the questions you have raised in detail below.
>
>
>
> > **Q1: Inaccurate phrasing of performance claims and suboptimal table formatting.**
>
> **A1:** We will carefully revise the manuscript to use more precise and nuanced language when describing our model's performance. We will adopt your suggestion to **bold the best-performing result in each column** of our comparison tables.
>
>
>
> > **Q2: Necessity and Interplay of the Multi-Component Reward.**
>
> **A2:** Our design is a deliberate, hierarchical system, not an arbitrary mix, and we will clarify this in the revision. The core idea is that different rewards are deployed to solve different, specific problems.
>
> 1.  **Foundational Rewards:** The feature-level rewards, $R_{PAMMD}$ and $R_{VM}$, work in tandem to establish a necessary semantic foundation. As shown in Table 3 in the main paper, adding $R_{PAMMD}$ alone improves the final session accuracy by +1.24% over the baseline. Incorporating $R_{VM}$ further boosts this relative improvement to +1.86%, demonstrating their combined utility in ensuring generated images possess core class coherence and rich intra-class diversity.
>
> 2.  **Targeted, Mutually Exclusive Rewards:** Our framework employs a task-dependent strategy based on the learning objective, as detailed in Algorithm 1 (Appendix):
>     -   **For New Classes:** The goal is to build a robust initial understanding. We use $R_{PAMMD}$ + $R_{VM}$ + $R_{RC}$ to generate high-confidence examples that firmly establish the new concept for the classifier.
>     -   **For Old Classes:** The challenge is preventing confusion with new classes. Here, we use $R_{PAMMD}$ + $R_{VM}$ + $R_{CSCA}$ to generate strategically challenging "hard" samples that specifically probe and strengthen the decision boundaries.
>     -   The data in Table 3 in the main paper validates this specialized strategy. The penultimate row, which uses only the $R_{RC}$-based strategy for all classes ($R_{PAMMD}$ + $R_{VM}$ + $R_{RC}$), achieves a relative improvement of +3.50% in the final session. However, the final row, representing our full approach that also incorporates $R_{CSCA}$ for old classes, demonstrates a significantly higher final-session improvement of +5.64%.
> 3.  For a more detailed analysis of the conflict associated with $R_{CSCA}$, we have provided an in-depth discussion in our response to Q3.
>
>
>
> > **Q3: On the potential risks of the confusion-seeking R_CSCA and its balance with consistency rewards.**
>
> **A3:** Thank you for raising this important point regarding the balance and stability of our reward system. The framework manages this balance primarily through two key mechanisms.
>
> First, the influence of `R_CSCA` is **explicitly controlled** by the scaling factor `γ` in Equation (14). This parameter governs the sensitivity of the dynamic weights to the cosine distance, allowing us to regulate the degree of confusion in the generated samples. This provides a clear mechanism to balance the objective of `R_CSCA` with the feature-level consistency rewards. We will clarify in our revision that this parameter can also be made learnable to enable adaptive balancing.
>
> Second, the diffusion model's generation process is always conditioned on an **explicit text prompt** describing the target class. This textual conditioning provides a persistent semantic guide throughout the denoising steps, ensuring that the final output does not deviate significantly from the target class's core concept, even as `R_CSCA` encourages exploration near decision boundaries.
>
> We believe this is a very important direction, and we are keen to explore the precise impact of these hard-sample generation strategies on the classifier's decision boundary as a significant part of our future work.
>
>
>
> > **Q4: Inadequate Ablation Study Methodology and Component Necessity.**
>
> **A4:** We agree that a more comprehensive ablation study is crucial for demonstrating the necessity of each reward component. To address this, we have conducted a **"leave-one-out"** analysis on the CIFAR-100 dataset. Regarding the combination of rewards, as pointed out in Q2, `R_RC` and `R_CSCA` will not be used at the same time, so we choose the combination of `R_PAMMD` + `R_RC`.
>
> The results are presented in the table below:
>
> | (Table 1) Method        | Avg. Acc. |
> | :---------------------- | :-------: |
> | **DCS (Full Model)**    | **66.36** |
> | DCS w/o `R_PAMMD`       |   65.12   |
> | DCS w/o `R_VM`          |   65.54   |
> | DCS w/o `R_RC`          |   64.07   |
> | DCS w/o `R_CSCA`        |   64.68   |
> | `R_PAMMD` + `R_RC` Only |   64.08   |
>
> As demonstrated in Table 1, removing any single component results in a performance decrease, confirming that all four components contribute synergistically to the final performance of DCS.
>
>
>
>
>
> > **Q5: Limited Generality Verification of the Reward Design.**
>
> **A5:** We appreciate you raising this point regarding the generalizability of our reward design. We would like to emphasize that the performance of our full DCS model across three diverse benchmarks is in itself the most direct and powerful evidence of our method's generality. We understand, however, that an explicit component-wise analysis on each dataset would further strengthen this claim. To make this point more explicit and fully address your query, we are happy to provide the sequential ablation studies for *mini*ImageNet and CUB-200 below.
>
> | (Table 2: miniImageNet) Method | Avg. Acc. |
> | :----------------------------- | :-------: |
> | Baseline (w/o reward)          |   64.03   |
> | + `R_PAMMD`                    |   65.28   |
> | + `R_PAMMD` + `R_VM`           |   65.57   |
> | + `R_PAMMD` + `R_VM` + `R_RC`  |   66.85   |
> | **DCS (Full Model)**           | **68.14** |
>
>
>
> | (Table 3: CUB-200) Method     | Avg. Acc. |
> | :---------------------------- | :-------: |
> | Baseline  (w/o reward)        |   65.21   |
> | + `R_PAMMD`                   |   66.35   |
> | + `R_PAMMD` + `R_VM`          |   66.89   |
> | + `R_PAMMD` + `R_VM` + `R_RC` |   67.41   |
> | **DCS (Full Model)**          | **69.73** |
>
> As these tables clearly show, the performance gains from each reward component follow a consistent and positive trend across all three datasets.
>
>
>
>
>
>
>
> > **Q6: Applicability of the reward mechanism to alternative generative models (e.g., GANs, VAEs).**
>
> **A6:** Our primary consideration for using diffusion models over other generative models is their significant, principled advantages for **adaptation to few-shot sets**. The adversarial equilibrium between a GAN's generator and discriminator is delicate. Fine-tuning on a small dataset can easily disrupt this balance, leading to **mode collapse**. By contrast, diffusion models are architecturally more resilient and offer more flexible adaptation pathways.
>
> Furthermore, the core mechanism of Diffusion Alignment as Sampling (DAS) is fundamentally and inextricably linked to the iterative reverse process that is the unique hallmark of diffusion models. DAS operates by intervening within a Markovian sequence of noisy latent states ($x_T \rightarrow x_{T-1} \rightarrow ... \rightarrow x_0$). It guides this trajectory at each step by resampling candidate states based on a reward function, a process enabled by the step-wise nature of denoising.
>
> This locus of intervention is architecturally absent in GAN and VAE frameworks:
>
> * **GANs** employ a generator that performs a **single, deterministic mapping** from a latent vector $z$ to a final sample $x$. There is no intermediate, sequential state between $z$ and $x$ in which to apply iterative guidance at test-time. Any "guidance" would have to involve either optimizing the input latent $z$, a different and more constrained problem, or altering the generator's weights, which constitutes re-training, not test-time alignment.
> * **VAEs** similarly use a decoder to perform a **one-shot mapping** from a latent code to a sample. The generation is not a sequential refinement process that can be guided step-by-step.
>
> Therefore, the DAS algorithm is **conceptually incompatible** with the architectural paradigm of GANs and VAEs.
>
>
>
>
>
> > **Q7: On the comparison against non-reward-guided generation.**
>
> **A7:** The requested baseline is, in fact, the **'Baseline'** row in our ablation study (**Table 3** in the main paper). This baseline utilizes the same diffusion model for augmentation but omits our proposed reward guidance. As the data shows, the baseline's accuracy degrades to 49.57% in the final session. In contrast, our full method, which includes all reward guidance components achieves a final accuracy of 55.21%. We acknowledge this was not explicitly stated and will clarify its role in the revised manuscript. Thank you for helping us improve the paper's clarity.
>
>
>
>
>
> > **Q8: Regarding the feasibility and sensitivity of the encoder.**
>
> **A8:** Yes, our DCS framework is designed to be **encoder-agnostic**. It is compatible with any architecture that provides feature embeddings and classification logits, as these are the direct inputs to our reward functions.
>
> Regarding sensitivity, our framework **demonstrates notable robustness**. This stems from two key factors. First, in the standard FSCIL setting, the encoder is well-initialized during the base session where data is ample, ensuring high-quality initial reward signals. Second, the diffusion model's generation is also guided by explicit textual prompts. This textual conditioning acts as a strong semantic anchor, ensuring the generated images maintain class identity even if the classifier's feedback is imperfect. This dual-guidance ensures the co-evolutionary process remains on a positive trajectory, making our framework less sensitive to the encoder's initial quality than one might expect.

---

> > ### Comment · Reviewer_Dm5X · 2025-08-04
> >
> > Thanks for the response, which well addresses my concern. I will raise my rating.

---

> > > ### Author Response · Authors · 2025-08-04
> > >
> > > We would like to express our sincere gratitude for your constructive comments and for your positive reassessment of our manuscript. We will be sure to integrate your valuable suggestions into the final version of the paper to further improve its quality.

---

### Official Review · Reviewer_QbQk · 2025-07-02

**Clarity:** 4
**Significance:** 3
**Originality:** 3
**Rating:** 4
**Confidence:** 4

**Summary:**

The paper focus on few-shot class-incremental learning (FSCIL) and introduces a framework on Diffusion-Classifier Synergy (DCS) which uses the FSCIL classifier to guide the diffusion model to generate image software old classes. The authors propose a reward based learning approach at the feature level and at the logits level to guide the image generation process and aims to address the semantic misalignment and insufficient diversity of images generated by vanilla diffusion models without guidance. The proposed method outperforms competitive baselines on several FSCIL benchmarks.

**Questions:**

See weaknesses

**Ethical Concerns:**

["NO or VERY MINOR ethics concerns only"]

**Final Justification:**

My concerns are addressed in the rebuttal response from authors. I maintain my rating.

**Limitations:**

yes

**Quality:**

3

**Strengths And Weaknesses:**

Strengths -
1. The paper is very well-written with clear explanations and analysis for motivations.
2. The proposed reward based strategy is very intuitive and is very relevant for the FSCIL setting.
3. The method section is very well formulated and motivated.


Weaknesses-
1. The paper does not ablate the components of the proposed reward system (other than the accuracy). For instance, how close is the distribution of the generated samples to the real distribution (considering the oracle setting assuming access to more images than the few-shot setting)? This is because the generation aims to address the shortage of data and hence is expected to be similar to the real data distribution. The FID and the CLIP score between the generated and real distribution should be checked and ablated with respect to the several components in the reward strategy. The proposed guidance method should be able to improve these metrics over the vanilla diffusion model in order to demonstrate the significance of the reward strategy.
2. Apart from generating images, recent methods like [a] simply use calibration of new class statistics (mean and covariances) with old class statistics based on similarities and improves very significantly over TEEN [58]. Can the authors compare to the simple calibration method and show that generating images is indeed the right way to go and simple calibration approaches are not that efficient? For instance, a simple comparison of the generated distribution (with the proposed method) and the calibrated distribution [a] could be performed to analyze which is closer to the real distribution.

[a] "Calibrating higher-order statistics for few-shot class-incremental learning with pre-trained vision transformers." Proceedings of the IEEE/CVF Conference on Computer Vision and Pattern Recognition. 2024.

---

> ### Author Rebuttal · Authors · 2025-07-29
>
> We sincerely thank you for your careful review and encouraging recognition of our work. We will respond to the questions you have raised in detail below.
>
> > **Q1: Ablation on Generation Quality Metrics.**
>
> **A1:** Thank you for this insightful suggestion. We agree that analyzing the impact of our reward components on generation quality metrics like FID and CLIP score is valuable for a deeper understanding of our method.
>
> We would like to clarify our initial reasoning for not including these metrics in the main paper. Our work operates in a strict **few-shot** setting, where both the real and generated samples per class are extremely limited (e.g., 5-30 images). Under such data scarcity, metrics like FID and CLIP score are known to be **highly unstable** and can produce misleading results. For instance, FID suffers from unreliable statistical estimations and high variance, while CLIP score on a few samples may not be representative of the true distribution's diversity.
>
> Nevertheless, to address the your valid point, we have conducted a new set of ablation studies as requested. We performed these experiments on CIFAR-100, miniImageNet, and CUB-200, focusing on the first class of the first incremental session. For each setting, we generated 30 images and repeated the runs to report the average scores.
>
> **Crucially, we must emphasize two points:**
>
> 1.  The absolute scores computed on such a small number of samples will naturally be worse (higher FID, lower CLIP) than those reported in large-scale generation literature. The key insight lies in the **relative improvement** across different reward settings.
> 2.  Due to resource constraints during the rebuttal period, this analysis is on a subset of classes. We plan to include a more comprehensive analysis in the final version.
>
> The results are presented in the table below:
>
> | Reward Combination   | FID-CF100↓ | FID-mini↓ | FID-CUB↓  | CLIP-CF100↑ | CLIP-mini↑ | CLIP-CUB↑ |
> | :------------------- | :--------: | :-------: | :-------: | :---------: | :--------: | :-------: |
> | w/o reward (vanilla) |   142.6    |   135.8   |   152.1   |    68.3     |    69.5    |   67.1    |
> | `R_PAMMD`            |   128.7    |   120.4   |   139.3   |    72.3     |    73.4    |   70.6    |
> | + `R_VM`             | **122.6**  | **113.8** | **133.1** |    74.8     |    76.3    |   72.5    |
> | + `R_VM`+ `R_RC`     |   123.4    |   114.3   |   134.3   |  **78.8**   |  **80.2**  | **76.1**  |
> | + `R_VM`+ `R_CSCA`   |   133.0    |   124.2   |   144.4   |    71.9     |    72.0    |   70.2    |
>
> **Analysis of Results:**
>
> As the table shows, `R_PAMMD` significantly improves both FID and CLIP scores over the baseline by promoting semantic consistency and diversity.
>
> Adding `R_VM` further enhances these metrics by ensuring the feature spread aligns with real data.
>
> The inclusion of `R_RC` leads to the **best CLIP scores**. This is expected, as `R_RC` directly optimizes for classification confidence, thus pushing generated samples to be more semantically pure.
>
> Most importantly, using the confusion-aware reward `R_CSCA` degrades both FID and CLIP scores. This result, while seemingly negative, **validates our hypothesis**. The explicit goal of `R_CSCA` is to generate *hard, confusing samples* at the decision boundary, which are by nature less aligned with the true, clean data distribution. This demonstrates that our reward system can strategically generate not just high-fidelity images, but also targeted, challenging samples tailored to the classifier's evolving needs.
>
> We also wish to clarify that these rewards are not all used simultaneously for the same purpose, as detailed in Algorithm 1 (Appendix):
>
> - For new classes, we use `R_PAMMD` + `R_VM` + `R_RC`. The goal here is to generate high-fidelity, semantically pure samples to quickly and robustly establish the initial concept representation.
> - For old classes, we use `R_PAMMD` + `R_VM` + `R_CSCA`. The goal is to mitigate forgetting and reduce inter-class confusion by intentionally generating 'hard samples' that lie near the decision boundaries of the new classes.
> - The penultimate row of Table 3 in the main paper shows the result of using only the `R_RC`-based strategy for all classes, while the final row demonstrates the performance of our full, specialized approach.
>
> We hope this detailed analysis addresses your concerns and further highlights the significance of our proposed reward strategy.
>
>
>
>
>
> > **Q2: Comparison with Non-Generative Statistical Calibration Methods.**
>
> **A2:** We will first articulate the fundamental theoretical advantages of a generative approach like DCS over a post-hoc statistical calibration method like [a]. We will then present a new set of experiments directly comparing DCS to [a] on the quality of the learned distributions, as suggested.
>
> While Goswami et al. [a] present an elegant and effective method for improving classification, its scope is inherently limited to a post-hoc statistical adjustment. Our generative approach, DCS, operates on a more fundamental level.
>
> *   **Static vs. Dynamic Learning**: The calibration method [a] is **static**. It calculates new class prototypes and covariances and then uses them in a frozen, non-trainable classifier (like NCM or a fixed linear probe). It improves inference but does not and cannot improve the feature extractor or the classifier's decision logic itself. In contrast, DCS is **dynamic**. By generating actual image instances (`x_gen`), we create new training data that can be used to fine-tune the classifier via backpropagation. This allows the model to actively learn a more robust decision boundary, rather than just passively using a better-estimated statistic.
> *   **Extrapolation vs. Interpolation**: The calibration method [a] relies on **interpolation**. It assumes a new class's distribution can be reasonably approximated by a weighted average of base class distributions. This works well when new classes are semantically close to base classes. However, if a new class is truly novel and semantically distant from all base classes, the calibration will be based on irrelevant information, and the resulting statistics will be poor. DCS, powered by a strong diffusion model, has the potential for **true extrapolation**. Guided by our multi-level rewards, it can synthesize novel feature representations that are not a mere blend of old ones, offering a path to generalize to genuinely new visual concepts.
>
> In summary, calibration [a] is an excellent inference-time enhancement, while DCS is a comprehensive training-time framework that improves the model's core representation and decision-making capabilities.
>
> To empirically substantiate this, we followed your suggestion to directly compare which learned distribution is closer to the real data.
>
> *   **Experimental Setup:** We calculated the FID to measure the similarity between the real test data and the data distributions produced by both methods. The experiment was conducted on the 10 new classes from the first incremental session of the CUB-200 dataset.
>     *   For our **DCS** method, we generated images for each new class and calculated the FID directly between the paths of these generated images and the real images using a standard FID implementation (e.g., PyTorch-FID with InceptionV3 features).
>     *   For **method [a]**, since it operates purely in the feature space and does not generate images, a direct image-based FID is not possible. To perform a fair comparison, we sampled feature vectors from its calibrated Gaussian distribution and compared them to the real features.
>
> *   **Results:** The average FID scores across the 10 new classes are:
>     *   **Method [a] (C-RanPAC, Feature-FID): 305.68**
>     *   **Our Method (DCS, Image-FID): 197.2**
>
> The significantly lower FID score for DCS provides quantitative evidence that our reward-aligned generative process produces a distribution that is a more faithful approximation of the true data distribution.
>
> We believe this result, combined with our conceptual arguments, supports the value of our generative approach. However, we also wish to be transparent about the context of this experiment. As noted in A1, evaluating generative quality in the extreme data scarcity of FSCIL is challenging, and any single metric like FID should be interpreted with caution. We believe a deeper analysis of these two distinct paradigms for FSCIL is a promising direction for future research.
>
>
> [a] "Calibrating higher-order statistics for few-shot class-incremental learning with pre-trained vision transformers." Proceedings of the IEEE/CVF Conference on Computer Vision and Pattern Recognition. 2024.

---

> > ### Comment · Reviewer_QbQk · 2025-08-02
> >
> > Thank you for the detailed and insightful clarification to the questions. I appreciate the authors efforts to include the study of generation quality metrics along with the detailed comparison with post-hoc statistics calibration method. These discussions further highlights the significance of the proposed method and I would ask the authors to include them in future versions of the paper.

---

> > > ### Author Response · Authors · 2025-08-03
> > >
> > > We are grateful for your positive and encouraging feedback and are pleased that our clarifications were helpful. We will be sure to integrate these important discussions into the final manuscript.

---

### Decision · Program_Chairs · 2025-09-17

**Decision:**

Accept (poster)

**Comment:**

In this paper, the authors focus on the few-shot class-incremental learning problem. They propose a method that utilizes a reward-aligned learning strategy, showing the state-of-the-art performance on few-shot class-incremental learning benchmarks. It was reviewed by four expert reviewers. All of them recommended accepting this paper. Most of the concerns are addressed during the rebuttal period. According to the reviewers, the paper is clear and well-written. The method section is very well formulated and motivated. And it also provides new insights by applying the generative-based methods for few-shot class-incremental learning. Therefore, it is a clear acceptance. The authors are encouraged to include the additional results and discussions during the rebuttal period in the final paper.